



# A novel injection technique:
# using a field-based quantum cascade laser for the analysis of gas samples derived from static chambers

Anne R. Wecking[1], Vanessa M. Cave[2], Lìyǐn L. Liáng[3], Aaron M. Wall[1], Jiafa Luo[2], David I. Campbell[1],
Louis A. Schipper[1]

[1] School of Science and Environmental Research Institute, The University of Waikato, Private Bag 3105,
 Hamilton 3240, Aotearoa New Zealand
[2] AgResearch Ruakura, Private Bag 3123, Hamilton 3240, Aotearoa New Zealand
[3] Manaaki Whenua – Landcare Research, Palmerston North 4442, Aotearoa New Zealand

*Correspondence to*: Anne R. Wecking (arw35@students.waikato.ac.nz), Louis A. Schipper (louis.schipper@waikato.ac.nz)

**Abstract.** The development of fast-response analysers for the measurement of nitrous oxide ($N_2O$) has resulted in exciting opportunities for new experimental techniques beyond commonly used static chambers and gas chromatography (GC) analysis. For example, quantum cascade laser absorption spectrometers (QCL) are now being used with eddy covariance (EC) or automated chambers. However, using a field-based QCL EC system to also quantify $N_2O$ concentrations in gas samples taken from static chambers has not yet been explored. Gas samples from static chambers are commonly analysed by GC that often requires labour and time consuming procedures off-site. Here, we developed a novel, field-based injection technique that allowed the use of a single QCL for: 1) micrometeorological EC; and 2) immediate manual injection of headspace samples taken from static chambers. To test this approach across a range of low to high $N_2O$ fluxes, we applied ammonium nitrate (AN) at 0, 300, 600 and 900 kg N ha$^{-1}$ ($AN_0$, $AN_{300}$, $AN_{600}$, $AN_{900}$) to plots on a pasture soil. After analysis, calculated $N_2O$ fluxes from QCL ($F_{N2O\_QCL}$) were compared with fluxes determined by a standard method, i.e. here laboratory-based GC ($F_{N2O\_GC}$). Subsequent comparison of QCL and GC derived data was tested using orthogonal regression, Bland Altman and bioequivalence statistics. For the AN treated plots, the mean cumulative $N_2O$ emissions across the seven day campaign were 0.97 ($AN_{300}$), 1.26 ($AN_{600}$) and 2.00 ($AN_{900}$) kg $N_2O$-N ha$^{-1}$ for $F_{N2O\_QCL}$ and 0.99 ($AN_{300}$), 1.31 ($AN_{600}$) and 2.03 ($AN_{900}$) kg $N_2O$-N ha$^{-1}$ for $F_{N2O\_GC}$. These $F_{N2O\_QCL}$ and $F_{N2O\_GC}$ were highly correlated (r = 0.996, n = 81) based on orthogonal regression, in agreement following the Bland Altman approach (i.e. within ± 1.96 standard deviations of the mean difference) and shown to be for all intents and purposes the same (i.e. bioequivalent). The $F_{N2O\_QCL}$ and $F_{N2O\_GC}$ derived under near-zero flux conditions ($AN_0$) were weakly correlated (r = 0.306, n = 27) and not found to agree or to be bioequivalent. This was likely caused by the calculation of small but apparent positive and negative $F_{N2O}$ when in fact the actual flux was zero, i.e. below the detection limit of static chambers. Our study demonstrated (1) that the capability of using one QCL to measure $N_2O$ at different scales, including manual injections, offered a great potential to advance field measurements of $N_2O$ (and other greenhouse gases) in future; and (2) that suitable statistics have to be adopted when formally assessing the agreement and difference (not only the correlation) between two methods of measurement.



## 1 Introduction

Accurate measurements of nitrous oxide ($N_2O$) emissions from agricultural land are crucial to quantify the contribution of the gas's radiative forcing to climate warming (Thompson et al., 2019). Nitrous oxide is a long-lived greenhouse gas with a global warming potential 265-times higher than that of carbon dioxide ($CO_2$) over 100 years, and is the largest contributor to the depletion of stratospheric ozone (IPCC, 2013; Ravishankara et al., 2009). Agricultural activities on intensively managed soils that receive high inputs of reactive nitrogen ($N_r$), mostly in the form of animal excreta and nitrogen fertiliser, are the main source of anthropogenic $N_2O$ emissions (Reay et al., 2012). Reactive nitrogen facilitates microbial nitrification and denitrification in the soil with $N_2O$ being an intermediate of these processes (Butterbach-Bahl et al., 2013; Firestone and Davidson, 1989). The production of $N_2O$ in soils is controlled by a multitude of environmental and anthropogenic factors, e.g. soil moisture, nitrogen input and overall farm management, which often result in highly variable $N_2O$ emissions (Erisman et al., 2013; Flechard et al., 2007; Rees et al., 2013). Adequate and precise flux measurements have, therefore, remained challenging (Cowan et al., 2020; Rapson and Dacres, 2014).

To date, the common method for measuring fluxes of $N_2O$ ($F_{N2O}$) are closed, non-steady-state 'static chambers' (Hutchinson and Mosier, 1981; Lundegard, 1927); a method used for more than 95 % of all field measurements (Lammirato et al., 2018; Rochette and Eriksen-Hamel, 2008; Rochette, 2011). Static chambers are relatively cost-efficient and easy to deploy in the field (de Klein et al., 2015; Velthof et al., 1996). Gas samples are extracted from the chamber headspace during an up to 60 min enclosure and injected into pre-evacuated glass vials (Luo et al., 2007; Rochette and Bertrand, 2003; van der Weerden et al., 2011). Subsequent analysis of the gas samples is commonly conducted off-site, using gas chromatography (GC) (Luo et al., 2008a; Parkin and Venterea, 2010). However, measurements using static chambers are discontinuous and labour-intensive with uncertainties in $F_{N2O}$ caused by alterations made to the soil environment after installation, pressure differences in the chamber headspace during sampling and the assumption of a linear increase/decrease in gas concentration with time (Chadwick et al., 2014; Christiansen et al., 2011; Denmead, 2008). Through time, different guidelines have been proposed to advance the standardisation of static chamber techniques (de Klein et al., 2015; Pavelka et al., 2018; Rochette, 2011) but essentially the basic method has remained unchanged for decades (Chadwick et al., 2014; Hutchinson and Mosier, 1981).

Alternative approaches to the static chamber method include the use of (semi-) automated chambers and micrometeorological techniques that allow $F_{N2O}$ measurements at higher temporal frequency and resolution (Baldocchi, 2014; Pavelka et al., 2018; Rapson and Dacres, 2014). Recent developments in the technology of fast-response analysers have enabled e.g. tunable diode laser absorption spectrometers, Fourier transform infrared spectrometers and, in particular, continuous-wave quantum cascade laser absorption spectrometers (QCL) to be coupled to automated chambers (Brümmer et al., 2017; Cowan et al., 2014; Savage et al., 2014) or eddy covariance (EC) systems (Nemitz et al., 2018; Nicolini et al., 2013). Despite these recent advances in analyser technology, our understanding of the micro- and macro-scale processes that lead to the emission of $N_2O$ has remained limited. While chamber measurements help to examine the interaction between soil processes and $F_{N2O}$ at point scales, EC promotes the understanding of diurnal, seasonal and annual $F_{N2O}$ dynamics at field to ecosystem scale (Cowan et al., 2020;



Liáng et al., 2018; Luo et al., 2017). Some studies have aligned chamber and EC measurements to determine the full range of processes that drive $F_{N2O}$ dynamics across these different scales but still relied on the use of more than one analyser for measuring $F_{N2O}$ (Jones et al., 2011; Tallec et al., 2019; Wecking et al., 2020a).

In this study, we tested whether a single field-deployed QCL could be used for manual injections of gas samples taken from static chambers to allow nearly concurrent measurements of chamber $N_2O$ samples alongside continuous EC. Field measurements using a QCL system for both these purposes have, to our knowledge, not yet been conducted. Our objective was to examine whether chamber $F_{N2O}$ determined by field-based QCL ($F_{N2O\_QCL}$) were equivalent to $F_{N2O}$ determined by laboratory GC ($F_{N2O\_GC}$). Orthogonal regression analysis was used to quantify the correlation between $F_{N2O\_QCL}$ and $F_{N2O\_GC}$. However, as correlation studies include limitations when assessing the comparability between two methods (i.e. GC and QCL), we also conducted Bland Altman and bioequivalence analyses to determined the degree to which $N_2O$ concentrations and fluxes derived from QCL would be comparable to GC.

## 2 Methods

### 2.1 Study site

This study was conducted at Troughton Farm, a commercially operating 199 ha dairy farm in the Waikato region, 3 km east of Waharoa (37.78°S, 175.80°E, 54 m a.s.l.), North Island, New Zealand. The farm had been under long-term grazing for at least 80 years with micrometeorological measurements using a QCL EC system made since November 2016 (Liáng et al., 2018; Wecking et al., 2020a). Mean annual temperature and precipitation, recorded at a climate station 13 km to the south-west of the farm (1981–2010), were 13.3 °C and 1249 mm, respectively (NIWA, 2018). The experimental site comprised three paddocks (P51, P53, P54) in the north of the farm with each sized about 2.8 ha. Soils were formed in rhyolitic and andesitic volcanic ash and rhyolitic alluvium. The dominant soil type based on the New Zealand soil taxonomy was a Mottled Orthic Allophanic soil (Te Puninga silt loam) (Hewitt, 2010). Plots used for the static chamber measurement of this study were located on P53 around 50 m to the south-west of the EC system.

### 2.2 Experiment design

One intensive field campaign was conducted between 10 and 16 September 2019. The campaign's primary purpose was to 1) manually collect gas samples from static chambers comprising potentially low to high $N_2O$ concentrations ($C_{N2O}$); 2) analyse these samples on-site using QCL and off-site using GC; 3) to quantify and compare resulting $C_{N2O}$ and $F_{N2O}$. A thorough description of the QCL operating in EC mode has been provided by Liáng et al. (2018) and Wecking et al. (2020a).

### 2.2.1 Static chamber measurements

The static chamber trial comprised a randomised block design of circular treatment and control plots each of which included three replicates per treatment/control. Ammonium nitrate (AN) fertiliser was used as a treatment and applied at different rates



to ensure production of a wide range of $F_{N2O}$ for subsequent flux measurements. The three applications rates were 300 ($AN_{300}$), 600 ($AN_{600}$) and 900 kg N ha$^{-1}$ ($AN_{900}$), while the control plots ($AN_0$) did not receive any AN. Separate areas adjacent to the twelve chamber plots were established to collect soil samples for laboratory analyses of soil moisture and soil mineral nitrogen ($N_{min}$). Soil moisture and water-filled pore space (WFPS) were analysed and calculated using the methods described in Wecking et al. (2020a). Soil $N_{min}$ was derived from field-moist soil samples extracted in 2M KCl (Mulvaney, 1996) and measured colorimetrically using a Skalar SAN++ flow analyser (Skalar Analytical B. V., Breda, Netherlands). Both, $NH_4^+$ and $NO_3^-$, were expressed in units kg ha$^{-1}$ using a site-specific soil dry bulk density of 0.73 g cm$^{-3}$ (Wecking et al., 2020a).

Flux measurements were made on the day of treatment application and throughout the following six days with chamber gas samples collected on nine occasions (Table S1). The sampling followed a standardised chamber technique (de Klein et al., 2003; de Klein et al., 2015; Luo et al., 2008b) and was carried out daily at 10 AM (NZDT) (van der Weerden et al., 2013). Additional sampling was conducted at noon on 12 and 15 September. Before sampling, PVC lids were fitted to water-filled base channels that provided a gas-tight seal over the 10 L headspace of the chambers. Gas samples were taken from this headspace during a 45 min enclosure period at four times – $t_0$, $t_{15}$, $t_{30}$ and $t_{45}$ – per chamber (Pavelka et al., 2018). A sampling port served to extract air from the chamber headspace by using a 60 mL plastic syringe (Terumo Corp., Tokyo, Japan). After flushing the syringe three times with air from the chamber headspace, the following procedure was applied to ensure that GC and QCL analysis received identical headspace samples: 1) after flushing, 60 mL of sample air was extracted from the chamber headspace; 2) 10 mL of the sample was discarded to flush the syringe needle; 3) 15 mL was transferred into a pre-evacuated, septum-sealed, screw-capped 5.6 mL glass vial (Exetainer, Labco Ltd., High Wycombe, UK); 4) the syringe needle was flushed again by discarding a further 10 mL; and 5) a second pre-evacuated glass vial was over-pressurised with 15 mL, and the remainder discarded. The procedure was repeated for each sample resulting in a total of 2 x 432 samples, i.e. two replicated sample batches for subsequent GC (1 x 432 samples) and QCL (1 x 432 samples) analyses.

### 2.2.2 Laboratory gas chromatography

Gas chromatography was conducted on the first sample batch at the New Zealand National Centre for Nitrous Oxide Measurements (NZ-NCNM) at Lincoln University, New Zealand. Automated analysis (GX-271 Liquid Handler, Gilson Inc., Middleton, WI) was performed using a SRI 8610 GC (SRI Instruments, Torrance, CA, USA) and a Shimadzu GC-17a (Shimadzu Corp., Kyoto, Japan) equipped with a $^{63}$Ni-electron capture detector. The analysis followed standard procedures described in detail by de Klein et al. (2015). Oxygen-free, ultra high purity nitrogen ($N_2$) was used as the carrier gas (mobile phase) at a flow rate of 0.4 L min$^{-1}$. The measurement frequency was set to 1 Hz. Sample exetainers experienced a storage time of up to two weeks prior to their analysis which was due to transportation from the field site to the laboratory. The run time during GC analysis was about eight minutes per sample.



### 2.2.3 Field quantum cascade laser absorption spectrometry

The second batch of $N_2O$ samples was analysed immediately after chamber sampling by manual injection into a continuous-wave quantum cascade laser absorption spectrometer (QCL, Aerodyne Research Inc., Billerica, MA, USA). Briefly, QCL uses infrared (IR) light energy which is passed through a 0.5 L multiple pass absorption cell with a pathlength of 76 m. Inside the cell, $N_2O$ absorbs IR light energy which then is quantified as equivalent to the compositional $N_2O$ concentration of the gas sample measured (Nelson et al., 2004).

For the purpose of our analysis, we switched the QCL from its continuous measurement (EC) mode to an 'injection mode'. The injection mode conversion took less than 30 minutes: a stainless steel three-way valve (Swagelok, Solon, OH, USA) mounted to the air inlet of the QCL allowed re-direction of the air flow from the primary inlet tube of the EC system into a second, 1 m long Bev-A-line tube (4 mm internal diameter). At its end, the tube was connected to a pressure regulator and a bottle of oxygen free, industrial grade $N_2$ carrier gas (BOC Ltd., NZ). Two stainless steel, T-junction connectors (Swagelok, Solon, OH, USA) were fitted to the sample tube allowing overflow of excess carrier gas through a 0.45 µm PTFE membrane filter (ThermoFisher, Scientific, NZ) and sample injection through a septum-sealed port (Fig. 1). A dry scroll vacuum pump (XDS35i, Edwards, West Sussex, UK) was used for both EC measurements and manual injections to continuously draw either air or carrier gas through the QCL sample cell.

Once the injection line had been established, the flow rate was reduced from an initial 15 L min$^{-1}$ used for EC to 1 L min$^{-1}$ for manual injections, based on Lebegue et al. (2016), Savage et al. (2014) and Brümmer et al. (2017). The reduction in flow was monitored using a RMA-SSV flow meter (Dwyer Instruments, PTY. Ltd., Michigan City, IN, USA) while setting the inlet control valve of the QCL to 2 V (using the TDLWintel software command) before manually adjusting inlet and outlet control valves of the QCL device further until the desired flow rate was achieved. Prior to sample injection, a minimum lag time of ten minutes was applied to let temperature and pressure of the QCL and its temperature-controlled enclosure box return to steady-state, i.e. $35 \pm 0.5$ Torr, 33.5 °C laser temperature and QCL enclosure box temperature of $30 \pm 0.1$ °C.

Standards of certified $N_2O$ concentration (range 0.2 to 100 ppm) were injected before, during and after each sample run and complemented QCL analysis (Table S2). Ten out of the twelve $N_2O$ standards were provided by the NZ-NCNM (except 0.321 and 0.401 ppm) and, therefore, were identical to those used for GC (Sect. 2.2.2). The QCL measurements were made at 10 Hz frequency with 1 mL of sample air extracted from each sample exetainer and manually injected into the flow of $N_2$ carrier gas by using a glass syringe (SGE International PTY Ltd., VIC, Australia). The glass syringe was flushed with $N_2$ gas after each injection to avoid cross-contamination of samples or $N_2O$ standards. Generally, using a 1 mL glass syringe was preferred to commonly used insulin syringes because of its higher accuracy resulting in greater output peak areas (Fig. S1c). The selection of syringe type, flow rate and the usage of $N_2O$ standards were based on preliminary tests conducted in advance of the actual field campaign (Fig. S1).



## 2.3 Data processing

Using GC and QCL resulted in the raw output of peak area data from injected $N_2O$ standards and chamber derived $N_2O$ samples. To compute final $C_{N2O}$, peak area data from $N_2O$ standards were fitted to linear and quadratic (second-order-polynomial) models (de Klein et al., 2015; van der Laan et al., 2009). Whereas de Klein et al. (2015) recommended the use of quadratic curves models for data measured by GC analysis, we found that both linear and quadratic models adequately fitted sample data derived from QCL. Using a linear fit ultimately resulted in on average 3 % smaller $F_{N2O\_QCL}$ (range -0.5 % to -4.3 %) than using a quadratic model. Nonetheless, since the quadratic fit suited lower $C_{N2O}$ better than a linear fit, quadratic models were preferred when generated from standards of known $N_2O$ concentration (Fig. S1a, b). The actual quadratic model used to calculate final $C_{N2O}$ of the gas samples was based on a selection of standards fitted to the expected minimum and maximum range of real sample $C_{N2O}$; which in our study did not exceed 10 ppm. Output data from GC were processed in PeakSimple software (SRI Instruments, Torrance, CA, USA) and Excel (Microsoft Corp. Redmond, WA, USA). MATLAB R2017a scripting (MathWorks Inc., Natick, MA, USA) was used for data derived from the QCL.

## 2.4 Flux calculation

The $N_2O$ flux in mg $N_2O$-N $m^{-2}$ $hr^{-1}$ was calculated for both data streams, GC ($F_{N2O\_GC}$, n = 108) and QCL ($F_{N2O\_QCL}$, n = 108), by applying a linear regression function to the increase in chamber headspace $C_{N2O}$ between time $t_0$ and $t_{45}$ following Eq (1) (van der Weerden et al., 2011):

$$F_{N2O\_GC} \text{ and } F_{N2O\_QCL} = \frac{\Delta N_2O}{\Delta T} \times \frac{M}{Vm} \times \frac{V}{A} \tag{1}$$

where $\Delta N_2O$ is the increase in headspace $C_{N2O}$ ($\mu L$ $N_2O$ $L^{-1}$ (ppmv)) over time; $\Delta T$ is the enclosure period (in hours); M is the molar weight of nitrogen in $N_2O$ (44 g $mol^{-1}$); Vm is the molar volume of gas (L $mol^{-1}$) at the mean air temperature recorded at each sampling occasion; V is the chamber headspace volume ($m^3$); and A is the area covered by the chamber base, here 0.0415 $m^2$. All $F_{N2O}$ were converted to units of nmol $N_2O$ $m^{-2}$ $s^{-1}$. The integration of $F_{N2O\_GC}$ (n = 84) and $F_{N2O\_QCL}$ (n = 84) determined at 10 AM sampling was used to quantify the proportion of applied nitrogen emitted as $N_2O$ ($E_{N2O}$) across the seven day trial in units kg $N_2O$-N $ha^{-1}$ based on Luo et al. (2007) and Wecking et al. (2020a).

## 2.5 Statistical analyses

The statistical analysis for $C_{N2O}$ data ($C_{N2O\_GC}$ and $C_{N2O}$_QCL, each n = 432) and resulting $F_{N2O}$ ($F_{N2O\_GC}$ and $F_{N2O\_QCL}$, each n = 108) was conducted in Genstat® (Version 19, VSN International, Hemel Hempstead, UK). After testing for normality using a Shapiro-Wilk test and homogeneity of variance by examining residual and fitted values, we applied three different statistical approaches to compare GC with QCL data: 1) orthogonal regression, 2) Bland Altman and 3) bioequivalence statistics.

The orthogonal regression analysis used standardised $C_{N2O}$ and $F_{N2O}$ data following Eq (2):

$$\text{standardised } C_{N2O} \text{ and } F_{N2O} = \frac{(x - mean)}{standard\ deviation} \tag{2}$$





The core of this orthogonal regression was a principal component analysis which, in contrast to ordinary least square regression, allowed for measurements errors in both the response and the predictor variable by minimising the squared residuals

in vertical and horizontal direction. While orthogonal regression returned a Pearson correlation coefficient r that provided information about the strength of the linear relationship between GC and QCL data, we found that r did not include any predication about the level of agreement between the two methods (Bland and Altman, 1986; Giavarina, 2015). The degree to which GC and QCL data would agree was, for that reason, determined by using Bland Altman statistics that quantified the bias (i.e. the mean difference) and the limits of agreement between the two methods. The limits of agreement were calculated

from the mean and the standard deviation (SD) of the difference between GC and QCL data. We defined that 95 % of all data points had to be within ± 1.96 SD of the mean difference (Giavarina, 2015). The Bland Altman analysis was conducted for individual $F_{N2O}$ as well as for mean $F_{N2O}$ across replicates of the same treatment.

Still, testing for correlation and agreement did not determine whether GC and QCL data would effectively and for practical purposes be the same (termed 'bioequivalent'). We, therefore, used bioequivalence statistics to assess the biological and

analytical relevance of the difference between the two methods. The first part of this analysis comprised an one-way analysis of variance (ANOVA) for $F_{N2O}$ which was subset by treatment ($AN_0$, $AN_{300}$, $AN_{600}$, $AN_{900}$) and analytical device (GC, QCL). Results from this ANOVA determined the 90 % confidence intervals (CI) of the mean difference between $F_{N2O\_QCL}$ and $F_{N2O\_GC}$. In bioequivalence statistics, the 90 % CI (corresponding to 80 % power) is generally preferred instead of using a 95 % CI that often serves to establish a statistical difference between two methods or treatments rather than proving no difference. An

important component of the analysis was to also define the bioequivalence range, i.e. the maximum acceptable difference, between the new (QCL) and the standard method (GC). Bioequivalence statistics acknowledge that two methods will never be exactly the same. Defining an acceptable bioequivalence range is, thus, an important precondition and might in some cases be even provided by a regulatory authority. While commonly used in pharmaceutical research (Bland and Altman, 1986; Giavarina, 2015; Patterson and Jones, 2006; Rani and Pargal, 2004), the concept of bioequivalence has not broadly been

applied in environmental sciences. Therefore, an acceptable bioequivalence range for $N_2O$ data based on the use of different analysers and methods has yet to be defined. We determined that the maximum acceptable difference of $F_{N2O\_QCL}$ in our study had to be as small as possible and within ± 5 % of the mean difference of the standard method ($F_{N2O\_GC}$). The null hypothesis ($F_{N2O\_QCL}$ is different from $F_{N2O\_GC}$) was rejected when the 90 % CI of the difference ($F_{N2O\_QCL}$-$F_{N2O\_GC}$) was entirely within the predefined bioequivalence range at a significance level of 5 %. Following the same principles, we conducted a

bioequivalence analysis for $C_{N2O\_QCL}$ and $C_{N2O\_GC}$.

## 3 Results and discussion

### 3.1 Environmental conditions and soil variables

Daily mean air temperatures during the seven-day chamber campaign ranged from 8.3 to 12.8 °C. The WFPS of chamber and associated soil plots did not fall below 73.9 % with a mean of 79.5 %. Cumulative rainfall for September 2019 was 119 mm





compared to only 2 mm occurring during the seven days of the campaign. As expected, soil $NH_4^+$ and $NO_3^-$ levels increased with increasing application of AN fertiliser. The highest values of $N_{min}$ measured at $AN_{900}$ plots were 265 kg $NH_4^+$ $ha^{-1}$ and 268 kg $NO_3^-$ $ha^{-1}$. The mean background levels of soil $NH_4^+$ and $NO_3^-$ were around 2 kg $ha^{-1}$. At the end of the campaign, soil $NH_4^+$ levels for all treatments had decreased by less than half while the amount of soil $NO_3^-$ remained similar to the initial level measured on the day of treatment application (Table S3).

### 3.2 Comparing GC and QCL derived data

#### 3.2.1 Magnitude and general variability

Measurements resulted in a wide range of $F_{N2O}$ but followed the same temporal and treatment-dependent patterns for both $F_{N2O\_GC}$ and $F_{N2O\_QCL}$. The magnitude of individual fluxes was between -0.10 and 22.24 nmol $N_2O$ $m^{-2}$ $s^{-1}$ for $F_{N2O\_GC}$ and -0.07 and 22.81 nmol $N_2O$ $m^{-2}$ $s^{-1}$ for $F_{N2O\_QCL}$. The mean $F_{N2O}$ (n = 27) from chamber plots that received the highest application

rate of AN fertiliser ($AN_{900}$) was 13.22 nmol $N_2O$ $m^{-2}$ $s^{-1}$ ± 1.47 (± standard error of the mean, SEM) for $F_{N2O\_GC}$ and 13.27 nmol $N_2O$ $m^{-2}$ $s^{-1}$ ± 1.43 for $F_{N2O\_QCL}$. Similarly, the $AN_{600}$ treatment had a mean $F_{N2O}$ of 8.51 nmol $N_2O$ $m^{-2}$ $s^{-1}$ ± 0.98 ($F_{N2O\_GC}$) and 8.33 nmol $N_2O$ $m^{-2}$ $s^{-1}$ ± 0.9 ($F_{N2O\_QCL}$). The mean $F_{N2O}$ for $AN_{300}$ was 6.61 nmol $N_2O$ $m^{-2}$ $s^{-1}$ ($F_{N2O\_GC}$) and 6.48 nmol $N_2O$ $m^{-2}$ $s^{-1}$ ± 0.69 ($F_{N2O\_QCL}$). At control plots, $F_{N2O}$ were close to zero (Fig 2; Table S3). We found that treatment $F_{N2O}$ increased from a near zero background flux to ≥ 8.5 nmol $N_2O$ $m^{-2}$ $s^{-1}$ on the second day of the campaign. From

then, $AN_{300}$ fluxes gradually decreased with time whereas $F_{N2O}$ for $AN_{600}$ and $AN_{900}$ remained relatively elevated until the last day of the trial (Fig. 2). These temporal trends align with Cowan et al. (2020) who observed $N_2O$ emissions to peak within seven days after urea and AN fertiliser application; and found that $F_{N2O}$ returned to background levels after two or three weeks. Similarly, short-term responses of $F_{N2O}$ to AN application were also determined by others, e.g. Bouwman et al. (2002); Jones et al. (2007) and Cardenas et al. (2019). However, for our study AN treatment effects on $F_{N2O}$ were of secondary interest.

Different rates of AN fertiliser were only applied to result in a wide range of $F_{N2O}$ (low to high) and thereby to allow for a methodological comparison of GC and QCL data.

#### 3.2.2 AN treatment flux and concentration data

The correlation between calculated $F_{N2O\_GC}$ and $F_{N2O\_QCL}$ and between $C_{N2O\_GC}$ and $C_{N2O\_QCL}$ across all treatments was high with an r value of 0.996 resulting from orthogonal regression (Fig. 3a, 3b). For both cases, major axis, ordinary and inverse

least squares were nearly identical to a 1:1 line. All three regression models could therefore be used similarly well to predict the strength of the linear relationship between $F_{N2O\_GC}$ and $F_{N2O\_QCL}$ and $C_{N2O\_GC}$ and $C_{N2O\_QCL}$, respectively (Table S4). The results of the orthogonal regression analysis suggested that QCL delivered equivalent data to the GC method. The Bland Altman statistic quantified a percentage difference between the two methods for $F_{N2O}$ (i.e. $F_{N2O\_GC}$ and $F_{N2O\_QCL}$ treatment means) of not smaller than -11.2 % and not greater than +9.2 % (Table S5). The percentage difference between individual

$F_{N2O\_GC}$ and $F_{N2O\_QCL}$ (not treatment means) was slightly greater but in only less than 3 % of all cases exceeded +10 % and -15


%, which was likely due to the higher variability of $F_{N2O}$ between individual replicates of the same treatment. For both cases, $\geq 95$ % of all data points were well within the pre-defined limits of agreement $\pm 1.96$ SD (Fig. 4b). The overall mean difference (bias) between $F_{N2O\_GC}$ and $F_{N2O\_QCL}$ was 0.1 nmol $N_2O$ m$^{-2}$ s$^{-1}$ (Fig. 4b). However, this small bias might be practically irrelevant when compared with the overall detection limit of static chambers and other general uncertainties. Neftel et al.

(2007), for instance, quantified a chamber detection limit of 0.23 nmol $N_2O$ m$^{-2}$ s$^{-1}$ whereas Parkin et al. (2012) reported 0.03 nmol $N_2O$ m$^{-2}$ s$^{-1}$. At the annual scale, Flechard et al. (2007) and others (e.g. Jones et al., 2011; Rochette and Eriksen-Hamel, 2008) showed that the uncertainty of integrated fluxes can be as high as 50 % when using the static chamber method.

### 3.2.3 Control flux and concentration data

In contrast to the strong comparability of GC and QCL data at AN treatment sites, $F_{N2O\_GC}$ and $F_{N2O\_QCL}$ measured at control

plots (AN$_0$) were only poorly correlated (r = 0.3064) (Fig. 3c). The model-fit of major axis, ordinary and inverse least squares indicated that the regression of $F_{N2O\_GC}$ on $F_{N2O\_QCL}$ (and vice versa) was not identical, i.e. differed in the minimisation of squared residuals in vertical and horizontal direction. Likewise, this also applied to $C_{N2O\_GC}$ and $C_{N2O\_QCL}$ (Fig. 3d). Mean $F_{N2O}$ ranged from a minimum of -0.05 to a maximum of only 0.21 nmol $N_2O$ m$^{-2}$ s$^{-1}$ (Table S3). Consequently, Bland Altman statistics determined only small quantitative differences between $F_{N2O\_GC}$ and $F_{N2O\_QCL}$. When computing the percentage

difference between these $F_{N2O\_GC}$ and $F_{N2O\_QCL}$, we found near-zero $F_{N2O}$ from AN$_0$ plots were less consistent in relative terms than treatment $F_{N2O}$ (Fig. 4, Table S5). However, these inconsistencies were generally small and did not appear of great biological interest.

More generally, QCL analysis resulted in slightly higher $C_{N2O}$ than GC, which might explain why the calculated $F_{N2O\_QCL}$ at AN$_0$ plots were higher than $F_{N2O\_GC}$ (Table S5). However, whether this finding was related to the potentially higher sensitivity

of the QCL device or due to other possible variations in sampling procedures was not resolved. Instead, we found that the disagreement between the GC and QCL method was likely related to ambient $N_2O$ concentrations in the chamber headspace that remained between 300-400 ppb and showed a non-linear response with time, regardless of which analytic device was used. This might have resulted in the calculation of very small but apparent positive and negative $F_{N2O}$, when in fact the actual flux was zero (*Type I error* as defined by Parkin et al. (2012)). The integration of $C_{N2O}$ with time to calculate $F_{N2O}$, therefore,

likely included this error; rather than being caused by uncertainties associated with measurement procedures or analytic device (Kroon et al., 2008). Hence, the deviation of $F_{N2O}$ determined at control sites (AN$_0$) from treatment $F_{N2O}$ (AN$_{300}$, AN$_{600}$, AN$_{900}$) has to be taken into account when evaluating the above results and mathematical principles (Sect. 3.2.2). Since static chamber measurements often include near-ambient $C_{N2O}$ and $F_{N2O}$ equal or near-zero, $F_{N2O}$ from control plots were kept in the manuscript for the purpose of completeness.

### 280    3.2.4 Cumulative N$_2$O emissions

Cumulative $N_2O$ emissions across the seven-day campaign were quantified slightly greater for the GC ($E_{N2O\_GC}$) than the QCL ($E_{N2O\_QCL}$) method. The mean difference between $E_{N2O\_GC}$ and $E_{N2O\_QCL}$ for the control (AN$_0$) and each treatment, AN$_{300}$, AN$_{600}$





and $AN_{900}$, was -0.011, +0.0023, +0.050 and +0.028 kg N ha$^{-1}$, respectively. This was a difference of less than 4 % in total $N_2O$ emissions during deployment (Fig. 5).

**3.3 Measurement performance of QCL analysis**

The measurement precision of QCL, and particularly GC, have been generally well-reviewed (de Klein et al., 2015; Lebegue et al., 2016; Rapson and Dacres, 2014). The precision of common GC analysers is < 0.5 ppb (Rapson and Dacres, 2014; van der Laan et al., 2009) while the precision of QCL was found to be about 0.3 ppb for measurements made at 10 Hz and 0.05 ppb for 1 Hz; but in some cases might be even higher (~1 ppt) (Curl et al., 2010; Rapson and Dacres, 2014; Savage et al.,
2014). Zellweger et al. (2019), for instance, used laboratory QCL for the calibration of $N_2O$ reference standards to inform the internationally accepted calibration scale of the Global Atmosphere Watch Programme of the World Meteorological Organisation. Similarly, Rosenstock et al. (2013) preferred lab-based QCL to verify the accuracy and precision of different photoacoustic spectrometers.

However, the analytic precision was also found to depend on factors other than the technical performance of the analytic
device. Rannik et al. (2015) indicated that the performance (and thus the precision of $F_{N2O}$) of an analyser to measure static chamber derived gas samples is likely more limited by the precision of the chamber system than by errors related to analysis or post-processing of the data itself. Imprecisions might be caused by several factors, e.g. chamber type and dimension, experimental set-up, deployment time and preferred sampling method, all of which would lead to differences in the flux detection limit (Sect. 3.2.2). In contrast, the sources of uncertainty in our study were most likely related to: 1) insufficient
evacuation of glass exetainers leading to the sporadic dilution of gas samples and $N_2O$ standards; and 2) variation of sample volume when injected into the QCL, which might not have been equal to 1 mL in practice and, thus, could have resulted in slight variations of output peak area. In agreement with these observations, de Klein et al. (2015) found that half the measurement uncertainty could be explained by the variability of gas sample volume in the sample exetainers. The inclusion of a fixed volume sample loop when injecting gas samples into the QCL might help to reduce this source of error.

As the $N_2O$ analysis using QCL was conducted in a temperature and pressure controlled environment, variations in these parameters were unlikely. The temperature dependency of $N_2O$ analysis by QCL was described as being linear by Lebegue et al. (2016) with variations less than 0.02 ppb °C$^{-1}$. To reduce the uncertainty of output peak area, we recommend a constant baseline flow of $N_2$ carrier gas at constant pressure (slightly higher than ambient) and temperature for manual injections made into the QCL device. Depending on the set-up of the QCL system, an initial lag time of 10 to 30 min before injections might
be required to ensure sufficient stabilisation of pressure and temperature in the QCL sample cell. Given a flow rate of 1 L min$^{-1}$, the delay between single injections of 1 mL sample volumes was short (5 to 8 sec). Sample concentrations at the same volume but > 20 ppm $N_2O$ required a longer delay time between individual injections (> 20 sec) to enable sufficient flushing of the QCL sample cell and to avoid cross-contamination. The identification of suitable delay times was straight forward and could be accessed easily in real time by visually examining the peak progression in TDLWintel. However, we did not determine
the extend to which spontaneous but small variations in the flow rate of $N_2$ carrier gas would have affected the resulting output





peak areas. Further uncertainties of true output peak areas might have also been associated with processing and curve fitting procedures applied to the raw dataset in MATLAB that likely led to small underestimations.

### 3.4 QCL injections

### 3.4.1 The concept of bioequivalence

Using the Pearson correlation coefficient and the coefficient of determination for comparing two or more quantitative methods is a generally preferred approach in the field of $N_2O$ research. Comparisons of different methods for $N_2O$ analysis made in the literature most commonly used orthogonal (Jones et al., 2011) and linear regression (Brümmer et al., 2017; Cowan et al., 2014; Tallec et al., 2019), Students t-tests (Christiansen et al., 2015) or were based on raw data (Savage et al., 2014). However, correlation studies as such have limitations when assessing the comparability between two methods since a correlation analysis

only identifies the relationship between two variables, not the difference (Giavarina, 2015). Bland Altman and bioequivalence statistics overcome this limitation by assessing the degree of agreement between methods.

An important aspect of statistical hypothesis testing is that the null hypothesis is never accepted. But failure to reject the null hypothesis is not the same as proving no difference. A bioequivalence assessment allows the statistical assessment of whether two methods (e.g. measurement devices, drug treatment) are effectively the same. Central to a bioequivalence analysis is the

330 "equivalence range" that defines the size of the acceptable difference for which the values are similar enough to be considered equivalent. This becomes important when considering that even with the most precise analytical design and the most tightly controlled experimental conditions, e.g. $F_{N2O\_GC}$ and $F_{N2O\_QCL}$ will never be exactly the same (Rani and Pargal, 2004). However, if the difference is sufficiently small for 'practical purposes', $F_{N2O\_GC}$ and $F_{N2O\_QCL}$ can be considered effectively the same. Here, an accepted proof of bioequivalence for $F_{N2O\_QCL}$ was if the 90 % confidence interval of the difference $F_{N2O\_QCL}$-$F_{N2O\_GC}$

(corresponding to a test with size 0.05) was within a ± 5 % difference of $F_{N2O\_GC}$.

The accepted proof of bioequivalence can vary depending on the objective of the research or guidelines provided by a regulatory authority but commonly does not exceed ± 20 % (Rani and Pargal, 2004; Ring et al., 2019; Westlake, 1988). In our study, a small bioequivalence range of ± 5 % was preferred to test the difference between $F_{N2O\_QCL}$ and $F_{N2O\_GC}$ since such recommendations did not exist.

Overall, our results showed that $F_{N2O\_GC}$ and $F_{N2O\_QCL}$ from $AN_{300}$, $AN_{600}$ and $AN_{900}$ plots met the criterion for bioequivalence. The 90 % confidence intervals of the difference ($F_{N2O\_GC}$-$F_{N2O\_QCL}$) were quantified 0.127 ($AN_{300}$), 0.185 ($AN_{600}$) and -0.043 ($AN_{900}$) nmol $N_2O$ m$^{-2}$ s$^{-1}$ and well within the pre-defined equivalence range of ± 5 % (Fig. 6e, Table S 6). At control sites ($AN_0$), $F_{N2O\_GC}$ and $F_{N2O\_QCL}$ did not meet the criterion for bioequivalence. However, the failure to establish bioequivalence for $AN_0$ sites was due to the overall limitation of the static chamber method to provide 'real' $F_{N2O}$; rather than based on failure

of the statistical principle (Sect. 3.2.3). On the contrary, when tested for $C_{N2O}$ instead of $F_{N2O}$, bioequivalence was confirmed for $t_0$ and $t_{15}$ but did not apply for $t_{30}$ and $t_{45}$ (Fig. 6a). Again, failure to establish bioequivalence was likely related to limitations of the static chamber method that, in this case, was indicated by the lower boundary of the 90 % CI remaining outside the





predefined bioequivalence ranges. Another possible reason for failing to establish bioequivalence for GC and QCL derived data at $AN_0$ sites could have been the maximum acceptable difference between the two methods itself. We defined (Sect. 2.5) that this difference had to be within $\pm 5$ % of the mean difference of the standard method (i.e. GC). However, it has to be taken into consideration that the accepted proof of bioequivalence would have led to different results if the percentage mean difference had been set to, for instance, $\pm 10$ %. Consequently, accepting a greater mean difference between the two methods would have resulted in determining bioequivalence for $C_{N2O\_GC}$ and $C_{N2O\_QCL}$ even at ambient concentration. More generally, we found that positive values of the 90 % CI of the difference indicated that the difference between the two methods (GC-QCL) resulted in higher $C_{N2O\_GC}$ and $F_{N2O\_GC}$. Negative values, instead, showed that the difference GC-QCL led $C_{N2O\_QCL}$ and $F_{N2O\_QCL}$ to be higher than $C_{N2O\_GC}$ and $F_{N2O\_GC}$. The overall difference between the two methods did not exceed $\pm 0.1$ ppm for $C_{N2O}$ and $\pm 0.38$ nmol $N_2O$ m$^{-2}$ s$^{-1}$ for $F_{N2O}$ (Fig. 6e).

To the best of our knowledge, bioequivalence has not broadly been applied in the greenhouse gas literature to identify and to discuss the range at which a difference in $F_{N2O\_GC}$ and $F_{N2O\_QCL}$ could be considered relevant when using different analytical methods. Defining the magnitude of $F_{N2O}$ (e.g. in nmol $N_2O$ m$^{-2}$ s$^{-1}$) at which a unit difference would actually become relevant, however, is important when using different methods to quantify, compare and ultimately upscale $N_2O$ emissions. We, therefore, recommend bioequivalence or other statistical approaches (e.g. Bland Altman) for more formally assessing the agreement between two methods in the future.

### 3.4.2 Strengths and weaknesses

The employment of a QCL analyser offers an alternative approach for the injection of $N_2O$ samples taken from static chambers, particularly as $F_{N2O\_QCL}$ were generally bioequivalent to $F_{N2O\_GC}$. Using a QCL for the purpose of manual injections can be conducted without much disruption to other measurements (e.g. EC or automated chambers) and, therefore, helps justify the initially higher capital and general running costs involved with operating a QCL device. Additional labour effort and time associated with sample storage and transport necessary for laboratory GC do not necessarily apply for field-based injections into a QCL. Once established, a QCL system has relatively low maintenance and offers a straightforward application for manual injections in addition to EC or other measurements. In our study, the assembly of the injection set-up required little equipment and was installed within 30 min. This allowed for a rapid analysis after chamber sampling without greatly interfering with other measurements, such as EC, that were offline during the time of manual injection into the QCL.

We acknowledge that sporadic dilution of $N_2O$ samples might have occurred for both GC and QCL analyses due to sample storage in and insufficient evacuation of sample exetainers (de Klein et al., 2015). Despite this potential source of uncertainty, storing $N_2O$ samples in exetainers enabled repeated injections from the same sample for multiple times and also allowed sample injections at suitable times, i.e. postponing analysis if EC measurements were of higher importance or if weather conditions (e.g. precipitation) did not support manual injections into the QCL (Faust and Liebig, 2018). Similar to GC, QCL injections required consumables ($N_2$ carrier gas and $N_2O$ standards) but, in contrast, time and costs associated with laboratory work were substantially less (Table 1).



## 4 Conclusion

Previously, QCL had been used either in conjunction with EC or coupled to automated chambers. Here, we showed that one QCL device could be used as a tool for the analysis of static chamber derived $N_2O$ samples without major disruption to these other measurement tasks. We found treatment $N_2O$ concentrations ($C_{N2O\_QCL}$) and fluxes ($F_{N2O\_QCL}$) from QCL were in

agreement with results based on laboratory GC ($C_{N2O\_GC}$, $F_{N2O\_GC}$). The percentage difference between treatment $F_{N2O\_GC}$ and $F_{N2O\_QCL}$ was not smaller than -11.2 % and not greater than +9.2 % with a mean difference between the two of only 0.1 nmol $N_2O$ $m^{-2}$ $s^{-1}$. Deviation between the GC and QCL methods was determined only for close to zero $F_{N2O}$ at control plots where $F_{N2O\_GC}$ and $F_{N2O\_QCL}$ values were found outside the predefined bioequivalence range. However, this was likely due to the calculation of very small but apparent positive and negative $F_{N2O}$ when in fact the actual flux was zero; rather than being caused

by uncertainties related to GC or QCL analysis itself. Bioequivalence was determined for all other $F_{N2O\_GC}$ and $F_{N2O\_QCL}$, i.e. it was confirmed that GC and QCL data were for practical purposes the same. We found that using Bland Altman and bioequivalence statistics in addition to regression analysis served the comparison of GC and QCL particularly well. Yet, these two statistical approaches have not been broadly used in the field of greenhouse gas research to compare different analytical methods or to discuss the magnitude at which a difference in $F_{N2O}$, or other greenhouse gas fluxes, would become relevant.

Since correlation studies identify the relationship between two methods but not the difference, we recommend that bioequivalence or other suitable statistical approaches are used for more formally assessing the agreement between two methods. Finally, QCL offers a great potential to interlink different methods of gas measurements across different temporal and spatial scales. In the future this capability might not only be important for measuring $N_2O$ but equally also applies to the measurement of other gas species (e.g. $CO_2$, $CH_4$) and gas isotopomers of interest.

**Data availability**

Data were deposited at the University of Waikato Research Commons, see (Wecking et al., 2020b) https://researchcommons.waikato.ac.nz/handle/10289/13539

**Supplements to this manuscript exist.**

**Author contributions**

ARW, VC, JL and LS designed the experiment. ARW performed the field work. ARW conducted the post-processing of GC and QCL data using MATLAB scripts provided by AW and DC. ARW performed the statistical analysis with inputs and contributions from VC. VC and LS commented on the results of the initial data analysis. ARW wrote and revised the manuscript with contributions from VC, AW, LL, JL, DC and LS.





**Competing interests**

The authors declare that they have no conflict of interest.

**Acknowledgements**

This research was supported by the New Zealand Agricultural Greenhouse Gas Research Centre (NZAGRC), AgResearch Ruakura, DairyNZ and the University of Waikato. The authors would like to recognise the farm owners, Sarah and Ben Troughton, for their cooperation. Chris Morcom is thanked for his help in the fields and Emily Huang from NZ-NCNM for

her all-embracing support regarding gas chromatography. Training notes on the concept of bioequivalence were gratefully received from Neil Cox. We would like to further acknowledge the continuous support from Aerodyne Research Ltd. in maintaining and advancing our QCL based EC systems. Finally, Cecile A. M. de Klein, Tom P. Moore and other (anonymous) reviewers are thanked for thoroughly revising the manuscript of this work.

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



**List of figures**

**Figure 1:** Schematic illustration of how to use a field-based QCL for EC measurements and manual injections. (1) shows the main components of the QCL EC system; (2) provides an example of a static chamber from which $N_2O$ samples were taken and stored in (3) pre-evacuated glass vials. Once the set-up for manual injections (4) was assembled and the QCL air-inlet (5) adjusted from drawing ambient air through the EC sample line (inlet 1) to drawing air through the injection tube (inlet 2), the QCL was readily set-up for receiving injections of $N_2O$ samples and associated standards through the injection port. The data output (6) was immediate allowing processing and data evaluation on the day of chamber sampling.

**Figure 2:** Fluxes of nitrous oxide ($F_{N2O}$) determined from (a) gas chromatography ($F_{N2O\_GC}$) and (b) quantum cascade laser absorption spectrometry ($F_{N2O\_QCL}$). Symbols depict mean $F_{N2O}$ and marker shading displays the rate of ammonium nitrate (AN) applied: $AN_0$ (black squares), $AN_{300}$ (dark grey diamonds), $AN_{600}$ (light grey upside-down triangles) and $AN_{900}$ (white triangles). Error bars illustrate the standard error of the mean (SEM) across the three replicates of the same treatment. Note that flux measurements on 12 and 15 September were conducted twice daily (10 AM and 12 PM) and that the time scale on the x-axis, therefore, is discrete. Soil water-filled pore space and mineral nitrogen contents associated with flux measurements are provided in the supplementary material, Table S3.

**Figure 3:** Orthogonal regression analysis of standardised $N_2O$ concentrations ($C_{N2O}$) and fluxes ($F_{N2O}$). Data were distinguished by their analytic source of origin, i.e. GC ($C_{N2O\_GC}$, $F_{N2O\_GC}$) and QCL ($C_{N2O\_QCL}$, $F_{N2O\_QCL}$). The regression analysis included all $C_{N2O}$ in (a) but only those $C_{N2O}$ measured at control sites ($AN_0$) in panel (c). The orthogonal regression analysis was repeated for standardised $F_{N2O}$ with (b) showing all $F_{N2O\_GC}$ and $F_{N2O\_QCL}$, and (d) depicting the orthogonal regression for $AN_0$ fluxes only. Ordinary least squares (dotted light grey line) resulted from the regression of Y on X; inverse least squares from the regression of X on Y (long dotted dark grey line). The major axis (black line) based on orthogonal regression of Y and X using a principal component analysis. Here, the squared residuals perpendicular to the line are minimised. Note, for the purpose of illustration axes in panel (c) and (d) have different scales. Table S4 in the supplements provides further results.

**Figure 4:** Bland Altman plots of the differences between the GC and QCL method expressed as the percentage difference of the standard method A ($F_{N2O\_GC}$) and the new method B ($F_{N2O\_QCL}$) on the y-axis [((A-B)/mean)× 100] versus the mean of A and B on the x-axis. The limits of agreement are represented by continuous lines at ±1.96 standard deviation (SD) of the percentage difference. The inset (panel b) illustrates the same data but excludes $F_{N2O\_GC}$ and $F_{N2O\_QCL}$ from control ($AN_0$) sites. The percentage mean difference (bias) between $F_{N2O\_GC}$ and $F_{N2O\_QCL}$, i.e. method A and B, is indicated by the gap between the dashed line (line of equality, which is not at zero) and an imaginary line parallel to the dashed line at y = 0. This figure is





based on individual $F_{N2O}$ (all treatment replicates). Results for mean $F_{N2O}$ across replicates of the same treatment are provided
in the supplements, see Table S5.

**Figure 5:** Cumulative emissions of $N_2O$ from each treatment ($AN_{300}$, $AN_{600}$, $AN_{900}$) and the control ($AN_0$) in kg $N_2O$-N ha$^{-1}$
at the end of the campaign. Data are distinguished into GC (black bars) and QCL (grey bars) budgets. Error bars quantify the
standard error of the mean (SEM). The absolute difference in kg $N_2O$-N ha$^{-1}$ between the two budgets (GC-QCL) is highlighted
by the number at the top of each bar-couple.

**Figure 6:** Bioequivalence analysis for $N_2O$ concentrations ($C_{N2O}$) in (a-d) and $N_2O$ fluxes ($F_{N2O}$) in (e) with GC defined as the
standard method. $C_{N2O}$ and $F_{N2O}$ based on QCL analysis were considered bioequivalent when the 90% confidence interval (CI)
of the difference between QCL and GC (x-axis) was completely within the predefined ± 5% bioequivalence range of the
625 difference of the standard method. The bioequivalence analysis was distinguished for $C_{N2O}$ by sampling interval ($t_0$, $t_{15}$, $t_{30}$,
$t_{45}$) and treatment with panel (a) showing results for control sites ($AN_0$) and panels (b), (c) and (d) for $AN_{300}$, $AN_{600}$ and $AN_{900}$
treatment sites. Similarly, a bioequivalence analysis was determined for $F_{N2O}$ in panel (e), here distinguished by AN application
rate on the y-axis.

**List of tables**

**Table 1:** Comparison of the GC and QCL injection methods. Details provided in the below table specifically relate to the
application of the two techniques in this study (i.e. have not been generalised). NZD = New Zealand dollars.





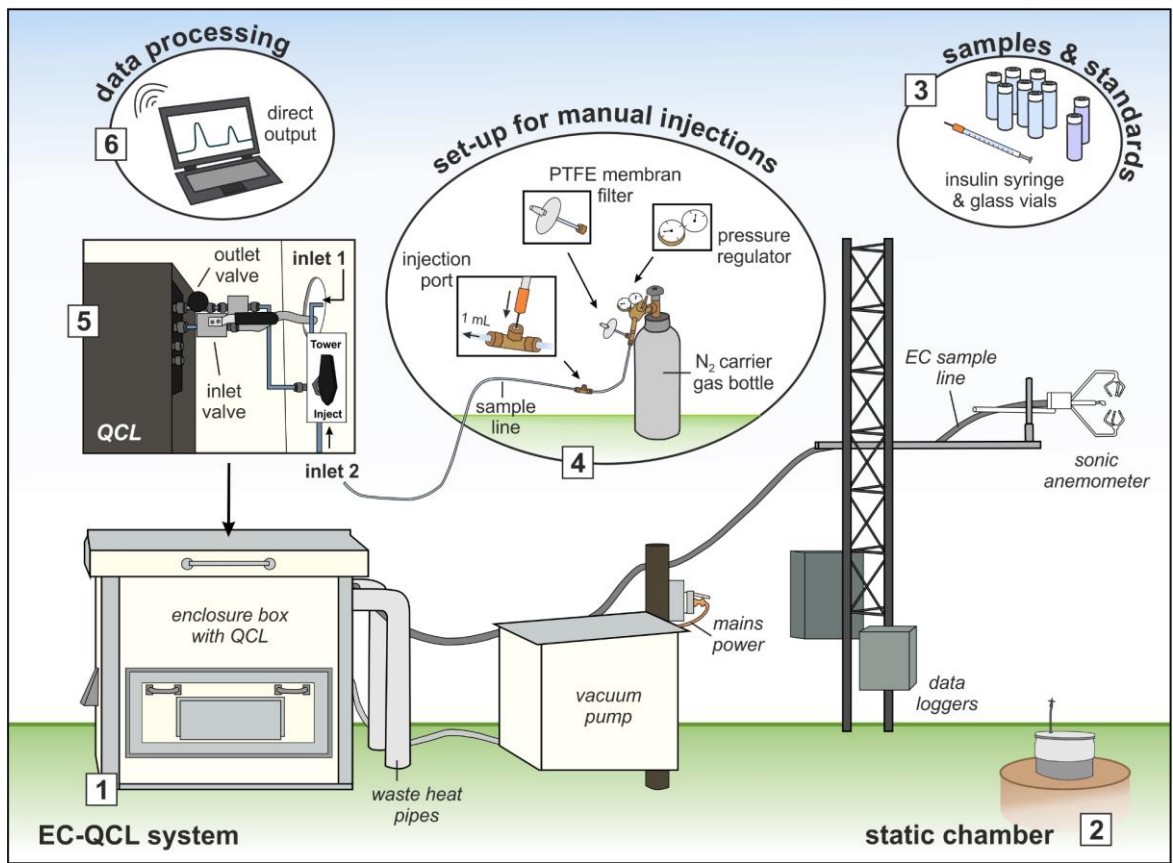

**Figure 1:** Schematic illustration of how to use a field-based QCL for EC measurements and manual injections. (1) shows the main components of the QCL EC system; (2) provides an example of a static chamber from which $N_2O$ samples were taken and stored in (3) pre-evacuated glass vials. Once the set-up for manual injections (4) was assembled and the QCL air-inlet (5) adjusted from drawing ambient air through the EC sample line (inlet 1) to drawing air through the injection tube (inlet 2), the QCL was readily set-up for receiving injections of $N_2O$ samples and associated standards through the injection port. The data output (6) was immediate allowing processing and data evaluation on the day of chamber sampling.





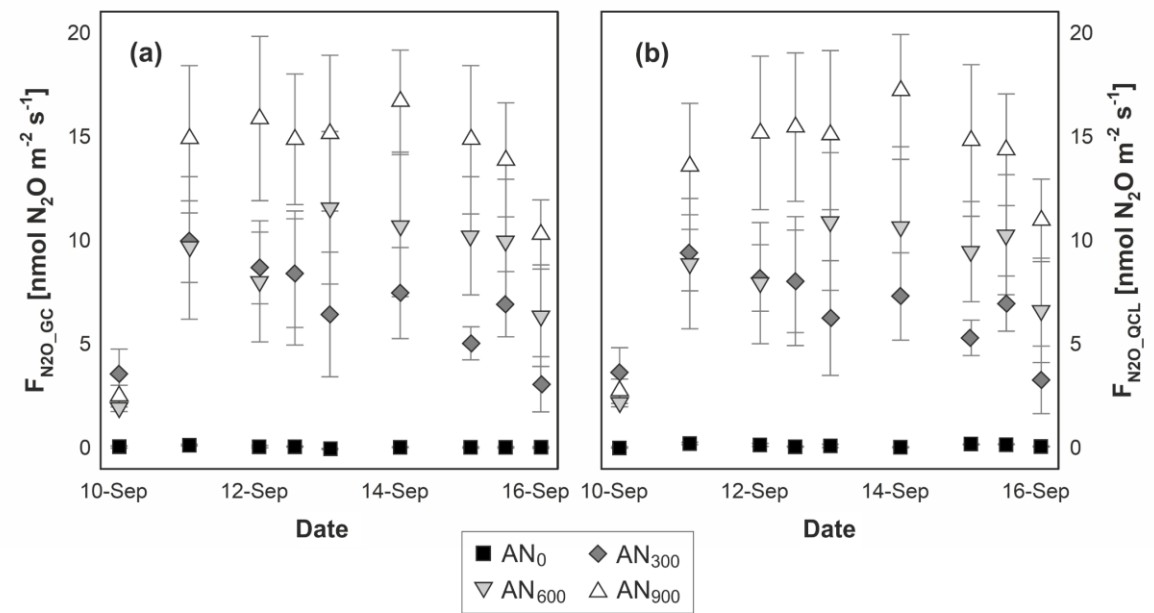

**Figure 2:** Fluxes of nitrous oxide ($F_{N2O}$) determined from (a) gas chromatography ($F_{N2O\_GC}$) and (b) quantum cascade laser absorption spectrometry ($F_{N2O\_QCL}$). Symbols depict mean $F_{N2O}$ and marker shading displays the rate of ammonium nitrate (AN) applied: $AN_0$ (black squares), $AN_{300}$ (dark grey diamonds), $AN_{600}$ (light grey upside-down triangles) and $AN_{900}$ (white triangles). Error bars illustrate the standard error of the mean (SEM) across the three replicates of the same treatment. Note that flux measurements on 12 and 15 September were conducted twice daily (10 AM and 12 PM) and that the time scale on the x-axis, therefore, is discrete. Soil water-filled pore space and mineral nitrogen contents associated with flux measurements are provided in the supplementary material, Table S3.



**Figure 3:** Orthogonal regression analysis of standardised $N_2O$ concentrations ($C_{N2O}$) and fluxes ($F_{N2O}$). Data were distinguished by their analytic source of origin, i.e. GC ($C_{N2O\_GC}$, $F_{N2O\_GC}$) and QCL ($C_{N2O\_QCL}$, $F_{N2O\_QCL}$). The regression analysis included all $C_{N2O}$ in (a) but only those $C_{N2O}$ measured at control sites ($AN_0$) in panel (c). The orthogonal regression analysis was repeated for standardised $F_{N2O}$ with (b) showing all $F_{N2O\_GC}$ and $F_{N2O\_QCL}$, and (d) depicting the orthogonal regression for $AN_0$ fluxes only. Ordinary least squares (dotted light grey line) resulted from the regression of Y on X; inverse least squares from the regression of X on Y (long dotted dark grey line). The major axis (black line) based on orthogonal regression of Y and X using a principal component analysis. Here, the squared residuals perpendicular to the line are minimised. Note, for the purpose of illustration axes in panel (c) and (d) have different scales. Table S4 in the supplements provides further results.



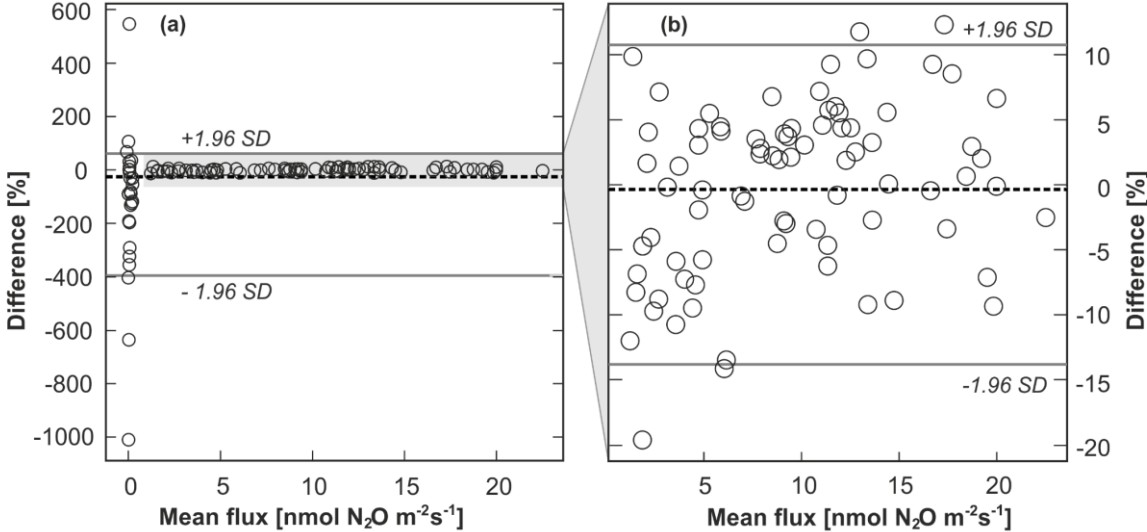

**Figure 4:** Bland Altman plots of the differences between the GC and QCL method expressed as the percentage difference of the standard method A ($F_{N2O\_GC}$) and the new method B ($F_{N2O\_QCL}$) on the y-axis [((A-B)/mean)× 100] versus the mean of A and B on the x-axis. The limits of agreement are represented by continuous lines at ±1.96 standard deviation (SD) of the percentage difference. The inset (panel b) illustrates the same data but excludes $F_{N2O\_GC}$ and $F_{N2O\_QCL}$ from control ($AN_0$) sites. The percentage mean difference (bias) between $F_{N2O\_GC}$ and $F_{N2O\_QCL}$, i.e. method A and B, is indicated by the gap between the dashed line (line of equality, which is not at zero) and an imaginary line parallel to the dashed line at y = 0. This figure is based on individual $F_{N2O}$ (all treatment replicates). Results for mean $F_{N2O}$ across replicates of the same treatment are provided in the supplements, see Table S5.





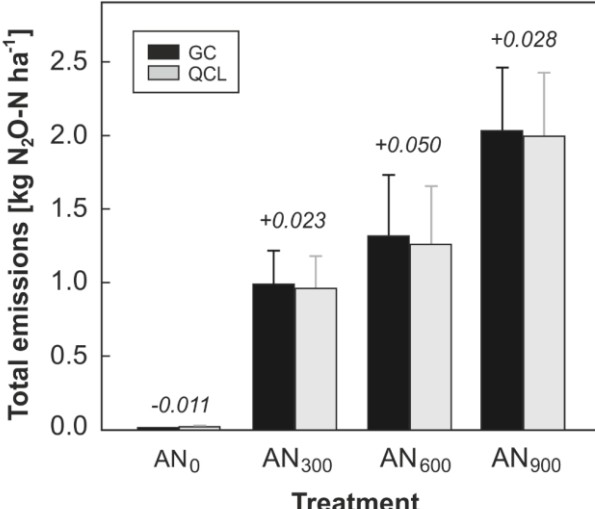

675

**Figure 5:** Cumulative emissions of $N_2O$ from each treatment ($AN_{300}$, $AN_{600}$, $AN_{900}$) and the control ($AN_0$) in kg $N_2O$-N ha$^{-1}$ at the end of the campaign. Data are distinguished into GC (black bars) and QCL (grey bars) budgets. Error bars quantify the standard error of the mean (SEM). The absolute difference in kg $N_2O$-N ha$^{-1}$ between the two budgets (GC-QCL) is highlighted by the number at the top of each bar-couple.

680









**Figure 6:** Bioequivalence analysis for $N_2O$ concentrations ($C_{N2O}$) in (a-d) and $N_2O$ fluxes ($F_{N2O}$) in (e) with GC defined as the standard method. $C_{N2O}$ and $F_{N2O}$ based on QCL analysis were considered bioequivalent when the 90% confidence interval (CI) of the difference between QCL and GC (x-axis) was completely within the predefined $\pm$ 5% bioequivalence range of the difference of the standard method. The bioequivalence analysis was distinguished for $C_{N2O}$ by sampling interval ($t_0$, $t_{15}$, $t_{30}$, $t_{45}$) and treatment with panel (a) showing results for control sites ($AN_0$) and panels (b), (c) and (d) for $AN_{300}$, $AN_{600}$ and $AN_{900}$ treatment sites. Similarly, a bioequivalence analysis was determined for $F_{N2O}$ in panel (e), here distinguished by AN application rate on the y-axis.

**Table 1:** Comparison of the GC and QCL injection methods. Details provided in the below table specifically relate to the application of the two techniques in this study (i.e. have not been generalised). NZD = New Zealand dollars.

|  | GC | QCL |
|---|---|---|
| Capital cost per device (NZD) | 40,000 | 160,000 |
| Labour effort for preparation and data processing of 100 samples (hours) | 2 to 3 | < 1 |
| Transport of samples | required | not required |
| Storage of samples | required | optional |
| Analysis location | lab-based | field-based |
| Analysis time (days) | multiple days | immediate |
| Analysis cost per sample (NZD) | 3.5 | < 0.5 |
| Possible injections (per hour) | 7.5 | ~200 |
| Lag time between injections (sec) | 480 | < 10 |
| Injection procedure | manual/automated | manual |
| Injection of $N_2O$ standards | required | required |
| Injection volume per sample (mL) | 6 | 1 |
| Carrier gas | $N_2$ | $N_2$ |
| Flow rate (L min$^{-1}$) | 0.4 | 1 |
| Output of result data | post analysis | immediate |