# Peer review of "using a field-based quantum cascade laser for the analysis of gas samples derived from static chambers"

_Atmospheric Measurement Techniques, 2020_

## Referee Comment (RC1) · Anonymous Referee #1 · 24 Jun 2020

The study "A novel injection technique: using a field-based quantum cascade laser for the analysis of gas samples derived from static chambers" by Wecking et al. describes an experiment in which samples of N2O measured using the static chamber method are then analysed on a QCL and GC instrument for comparison. The study is well written and presented, but there are some over simplifications that should be addressed in how the complexity of the system is described and the way the data is handled in the study. I advise quite a significant re-write focussing on the actual focus of the study, which is how the instruments compare. My first comment is that this study is essentially a comparison of concentrations measured using two instruments. This work could have been carried out with gas standards without the need for any chamber

measurements. The real point of the study here is whether gas injected through a QCL following this setup is a valid way to measure gas concentrations. If so, then fluxes calculated from the sample will compare well regardless. The novelty of the method is that the authors get past the inability of the QCL to measure actual concentrations of a sample by using standards to integrate peak areas, similar to the way the GC reports measurements, thus reducing the sample volume required. I'd like to see some examples of the QCL concentration output at 10Hz while measuring low, medium and high concentrations of N2O in the standards to observe the shape of the peaks that are integrated. I think the presentation of the integration of these peaks far outweighs the flux work. By the authors own numbers, I believe a flush rate of the cell is greater than 1 second, so I'd be interested to see how the laser reports concentrations while measuring at 10Hz, and what the noise looks like. As this is the real novelty in the manuscript, much more emphasis should be on the outputs of the instrument itself, and less on the flux measurements. I agree with many of the points the authors make, and I feel the study has value as a reference for people who may want to use the method presented in future studies; however, I don't see it being popular. Most Eddy C sites will value the collection of uninterrupted data over the ability to run the instrument as a makeshift GC. Choosing manual injections over the more common auto injection systems used by the GC also introduce a bit of a time cost and add room for human error. In terms of running costs, a QCL requires an air conditioned site with mains power to operate in EC mode, which is essentially a lab in itself. This system is expensive to run in terms of power and replacement of parts (pump maintenance, laser lifetime of approx. 7 years etc...). The author's point is that this system is already up and running, so the additional ability to do chamber measurements doesn't add cost. This is true, and although it's also true to say you don't need to take samples off-site, you will still likely have to travel some distance as plot experiments really shouldn't be setup within the footprint range of an eddy C system (ideally more than 1 km away due to the exponential rise in fluxes observed after N fertiliser application and the potential for advection effects which nullify the assumptions made by the eddy c method). The air conditioned site

thus becomes a mini-lab, in some cases closer to a field site where chamber measurements are made. Some discussion on the limitations of use is required, as the study seems to suggest N fertiliser plots could be setup next to an EC tower which would be very unwise unless the plots were to mimic the exact conditions of the field of interest to the EC measurements. In terms of the flux experiment, I find the application of fertilisers to be far larger than is common practice. 300 Kg N ha-1 is very large, and 600 and 900 is beyond realistic. In these cases I assume some kind of saturation of N in the soil and N2O in the chamber during a 45 minute enclosure which would also affect the magnitude of the fluxes observed. In any case, the fluxes reported are of little use other than to compare the instruments. In this case, there is no reason to take means from plots. Due to the log-normal nature of N2O emissions, the (arithmetically derived) mean values reported from a small n size (less than 25 chambers) is going to be fairly uncertain. Without accounting for the log-normal nature of these fluxes in both time and space, any uncertainties in cumulative flux estimates are not statistically meaningful. I return to my original point that this comparison is of gas concentrations and not of plots. The fluxes derived from both instruments are valuable on a 1:1 basis as presented in Figs 3a and 3b. That's all the paper requires and it's a great result in terms of showing the system works as well as the GC. In conclusion, I think the work presented is a well carried out and valid study, but it needs a bit of a re-write to focus on the actual message, and not get distracted by flux comparisons and methods of comparing significance.

---

## Author Comment (AC1) · 30 Jun 2020

The study "A novel injection technique: using a field-based quantum cascade laser for the analysis of gas samples derived from static chambers" by Wecking et al. describes an experiment in which samples of N2O measured using the static chamber method are then analysed on a QCL and GC instrument for comparison. The study is well written and presented, but there are some over simplifications that should be addressed, in how the complexity of the system is described and the way the data is handled in the study. I advise quite a significant re-write focussing on the actual focus of the study, which is how the instruments compare.

[Figure]

* We like to thank Referee #1 for her/his extensive comments. Our response to these comments is indicated with an asterix (*) at the beginning and the end:

For clarification, the interest of our study was not only to compare two different analytic devices but to further the suitability of a field-based quantum cascade laser (QCL) and associated sampling/measurement procedures to measure chamber derived samples of nitrous oxide (N2O). Of particular interest for us was to develop a technique that is applicable and easy to conduct in field environments, and that can be used in conjunction with eddy covariance (EC). Furthermore, we compared resource efficiency of the whole operation which is critical for decisions that will be made by researchers when considering actual costs of different research approaches and analytical devices. *

My first comment is that this study is essentially a comparison of concentrations measured using two instruments. This work could have been carried out with gas standards without the need for any chamber measurements. The real point of the study here is whether gas injected through a QCL following this setup is a valid way to measure gas concentrations. If so, then fluxes calculated from the sample will compare well regardless.

* Using gas standards would have allowed for a comparison between GC and QCL. However, this would not have included the whole sampling/analysis approach. Initial lab-based comparisons of QCL to other analytic devices have already been provided in the literature and shown that the suitability of a QCL device to quantify N2O standards is without doubt (Zellweger et al. (2019); Rosenstock et al. (2013) ). We were, therefore, highly interested in whether the lab-based accuracy and precision of a QCL could also be achieved in a field-based "mini-lab" – and under realistic sampling procedures that included N2O sampling from static chambers, sample separation to inject identical sample volumes and concentrations into either of the two analysers, sample storage, sample and data processing. *

The novelty of the method is that the authors get past the inability of the QCL to measure actual concentrations of a sample by using standards to integrate peak areas, similar to the way the GC reports measurements, thus reducing the sample volume required.

* Agreed, this is one advantage of using the QCL in the field setting. Further benefits were 1) the process of getting to the point of receiving these peak area data (e.g. developing sampling procedures and the injection set-up in a field environment); and 2) to test whether the QCL could reliably process samples of low, medium and high N2O concentration (that were derived from static chambers in a real scenario including method specific uncertainties). *

I'd like to see some examples of the QCL concentration output at 10Hz while measuring low, medium and high concentrations of N2O in the standards to observe the shape of the peaks that are integrated. I think the presentation of the integration of these peaks far outweighs the flux work. By the authors own numbers, I believe a flush rate of the cell is greater than 1 second, so I'd be interested to see how the laser reports concentrations while measuring at 10 Hz, and what the noise looks like.

* L. 150: All our injections were conducted at a 10 Hz frequency, which means that all QCL concentration data reported in the manuscript were consistently determined at this particular frequency. Examples of how low to high N2O concentrations were reported by the QCL were provided in the supplementary material of the work, i.e. Figure S1 a) and b). Raw data can also be found in the data repository associated with the manuscript. However, we are in line with the Referee that including a raw 10 Hz time series of injection peaks might be a useful addition to the manuscript/the supplementary material and will be considered. *

As this is the real novelty in the manuscript, much more emphasis should be on the outputs of the instrument itself, and less on the flux measurements.

* The focus of our manuscript is on the concentration output of the QCL and GC analysers as represented by the majority of section 2 and section 3.2, 3.3, 3.4. Calculating

the resulting N2O fluxes was a necessary exercise in addition to the emphasis on N2O concentrations. We like to point the Referee to the objective of the bioequivalence analysis applied to both N2O concentration and flux data (see Section 3.4.1). Determining the suitability of a measurement device for a particular purpose has to be evaluated. However, a simple comparison of concentration data based on linear regression does not satisfy this purpose. Consequently, we were led by the intention to not only assess the degree to which the two methods (GC and QCL) would agree (orthogonal regression, Bland Altman). But to further determine whether N2O concentration and flux data, in fact, were statistically speaking "the same" (i.e. bioequivalent). Receiving reliable N2O fluxes when using the QCL injection technique was the ultimate goal of our study. We like to acknowledge that applying statistical tests like bioequivalence describes a concept most N2O flux researchers might be unfamiliar with. However, to assess the comparability of data derived from two different methods these tests are essential as was our evaluation of N2O flux data to discuss the relevance of our results in a real world scenario. *

I agree with many of the points the authors make, and I feel the study has value as a reference for people who may want to use the method presented in future studies; however, I don't see it being popular. Most Eddy C sites will value the collection of uninterrupted data over the ability to run the instrument as a makeshift GC.

* It is good that we have general agreement about the value of the work. One of the key purposes of this work was to provide information for operators to make informed decisions about how they might operate their eddy covariance sites and consider trade-offs of data loss for short periods of time. How popular this approach becomes remains to be seen but was not really a focus of the current study. Thus, even if not prominent at present, we are convinced that using a QCL in the most cost- and time-efficient way to suit multiple research purposes provides a promising future application. *

Choosing manual injections over the more common auto injection systems used by the GC also introduce a bit of a time cost and add room for human error.

[Figure]

\* This is correct and this human error is captured within the error reported in the paper. Sample loops could be included in our proposed setup to further reduce the error, if desired. The time taken for one injection was less than 10 seconds and accounted for in our time analysis (Table S2) which shows overall a large time saving of, in our case, multiple days (considering transport of samples to the lab and analysis time, see Table 1). We could show that at a given flow rate of 1 L min-1 the delay time between single injections of 1 mL sample volumes was generally short (5 to 8 sec). The return of the N2O concentration in the QCL sample cell to basically zero (= background level of the N2 carrier gas) was straight forward when using visual examination of the real-time curve in TDL-Wintel. Suitable delay times were, thus, easily adjustable if the sample concentration would have exceeded 20 ppm N2O (line 312). In our study, field samples did not exceed a N2O concentration greater 10 ppm. This supported the applicability of our approach. It is to consider that using an auto-injection system would have to account for different delay times after sample injection and, therefore, would likely include longer sample delay and overall sample analysis times. \*

In terms of running costs, a QCL requires an air conditioned site with mains power to operate in EC mode, which is essentially a lab in itself. This system is expensive to run in terms of power and replacement of parts (pump maintenance, laser lifetime of approx. 7 years etc.). The author's point is that this system is already up and running, so the additional ability to do chamber measurements doesn't add cost.

\* We agree. \*

This is true, and although it's also true to say you don't need to take samples off-site, you will still likely have to travel some distance as plot experiments really shouldn't be setup within the footprint range of an eddy C system (ideally more than 1 km away due to the exponential rise in fluxes observed after N fertiliser application and the potential for advection effects which nullify the assumptions made by the eddy c method). The air conditioned site thus becomes a mini-lab, in some cases closer to a field site where chamber measurements are made. Some discussion on the limitations of use is required, as the study seems to suggest N fertiliser plots could be setup next to an EC tower which would be very unwise unless the plots were to mimic the exact conditions of the field of interest to the EC measurements.

* We positioned and aligned all static chamber measurements with overall site management conditions and associated research goals. Chambers were placed outside the immediate footprint of the eddy flux tower as determined by Wall et al. (2020). This meant that the chamber trial area was about 100 m (walking distance from the tower) to avoid the Referee's concerns. Our previous work (Wecking et al., 2020), provided an example of how to conduct chamber measurements near to an EC tower. We demonstrated that the flux signal originating from the chamber plots was not strong enough to impact our EC measurements. To address the reviewers concern, we are going to add in a sentence recommending that care has to be taken when locating chambers near an EC tower to avoid cross-contamination. *

In terms of the flux experiment, I find the application of fertilisers to be far larger than is common practice. 300 Kg N ha-1 is very large, and 600 and 900 is beyond realistic. In these cases I assume some kind of saturation of N in the soil and N2O in the chamber during a 45 minute enclosure which would also affect the magnitude of the fluxes observed. In any case, the fluxes reported are of little use other than to compare the instruments.

* Applications rates of ammonium nitrate fertiliser were chosen intentionally to trigger different low, medium and high N2O fluxes (line 18, line 95 f.). In this study, we did not intend to mimic realistic fertiliser application scenarios. Having said this, however, the N deposited in a single urine patch of dairy cattle (i.e. a major source of N2O emissions) are comparable to N loading commonly observed at 600 kg N2O-N ha-1 (Selbie et al., 2015, p. 238 and references therein). *

In this case, there is no reason to take means from plots. Due to the log-normal nature of N2O emissions, the (arithmetically derived) mean values reported from a small n size

(less than 25 chambers) is going to be fairly uncertain. Without accounting for the log-normal nature of these fluxes in both time and space, any uncertainties in cumulative flux estimates are not statistically meaningful.

* The study's intention was not to discuss the effect of different applications rates of ammonium nitrate fertiliser on N2O fluxes. Treatment effects were only of secondary interest. Different rates of fertiliser were only applied to result in a wide range of N2O fluxes (low to high), and thereby to allow for a methodological comparison of GC and QCL data. *

I return to my original point that this comparison is of gas concentrations and not of plots. The fluxes derived from both instruments are valuable on a 1:1 basis as presented in Figs 3a and 3b. That's all the paper requires and it's a great result in terms of showing the system works as well as the GC.

* Our study is a comparison of two methods, GC and QCL. Validation of this comparison is retrieved from different statistical tests (orthogonal regression, Bland Altman and bioequivalence) that enabled us to use a QCL for the purpose of analysing static chamber derived N2O samples under real field conditions. N2O concentration and flux data were determined and used to verify this objective. *

In conclusion, I think the work presented is a well carried out and valid study, but it needs a bit of a re-write to focus on the actual message, and not get distracted by flux comparisons and methods of comparing significance.

* Perhaps this is where we disagree, we are focussed on comparing the whole process required to quantify fluxes from a chamber trial using a field-based QCL rather than a laboratory based GC. Even a field based GC would give a very different outcome because of the longer time needed for separating and analysing samples. Our study will provide a broader assessment for researchers considering using QCLs in the field including quantifying precision and accuracy of the approach coupled to time and resource needs. In addition, testing the suitability of a new method (QCL) compared with

a standard method (GC) requires suitable statistical tests that do not only compare the significance but let us evaluate the agreement and bioequivalence of these methods (line 198, we also recommend reading the references quoted in Section 2.5 – Bland and Altman, 1986 and Giavarina, 2015 – that provide powerful insights into the warrant of applying these statistical tests to both N2O concentrations and fluxes in our study). *

Yours sincerely, A. Wecking in the name of all associated Co-Authors Hamilton, New Zealand – 30-06-2020

BLAND, J. M. & ALTMAN, D. G. 1986. Statistical methods for assessing agreement between two methods of clinical measurement. The Lancet, 327, 307-310.

GIAVARINA, D. 2015. Understanding Bland Altman analysis. Biochemia Medica, 25, 141-151.

ROSENSTOCK, T. S., DIAZ-PINES, E., ZUAZO, P., JORDAN, G., PREDOTOVA, M., MUTUO, P., ABWANDA, S., THIONG'O, M., BUERKERT, A., RUFINO, M. C., KIESE, R., NEUFELDT, H. & BUTTERBACH-BAHL, K. 2013. Accuracy and precision of photoacoustic spectroscopy not guaranteed. Global Change Biology, 19, 3565-3567.

SELBIE, D. R., BUCKTHOUGHT, L. E. & SHEPHERD, M. A. 2015. Chapter Four - The Challenge of the Urine Patch for Managing Nitrogen in Grazed Pasture Systems. Adv. Agron., 129, 229-292.

WALL, A. M., CAMPBELL, D. I., MUDGE, P. L. & SCHIPPER, L. A. 2020. Temperate grazed grassland carbon balances for two adjacent paddocks determined separately from one eddy covariance system. Agricultural and Forest Meteorology, 287, 107942.

WECKING, A. R., WALL, A. M., LIÁNG, L. L., LINDSEY, S. B., LUO, J., CAMPBELL, D. I. & SCHIPPER, L. A. 2020. Reconciling annual nitrous oxide emissions of an intensively grazed dairy pasture determined by eddy covariance and emission factors. Agriculture, Ecosystems & Environment, 287, 106646.

[Figure]

ZELLWEGER, C., STEINBRECHER, R., LAURENT, O., LEE, H., KIM, S., EMMENEG-GER, L., STEINBACHER, M. & BUCHMANN, B. 2019. Recent advances in measurement techniques for atmospheric carbon monoxide and nitrous oxide observations. Atmos. Meas. Tech., 12, 5863-5878.

———————————————————

---

## Referee Comment (RC2) · Anonymous Referee #2 · 24 Jul 2020

In their submission "A novel injection technique: using a field-based quantum cascade laser for the analysis of gas samples derived from static chambers" Wecking et al. report on their experiences with a sampling strategy that analyses both eddy covariance data and chamber measurements on-site using the same insrument. They calculated fluxes of N2O from concentrations measured by GC (conventional) and directly injected into a QCL (novel) in three different fertilisation treatments (+ control). Both concentrations anf fluxes were analysed with a variety of statistical methods and found to be practically equivalent, with the exception of near-zero flux conditions.

I have thoroughly enjoyed reviewing this manuscript. It is well-structured, well-written,

and the figures and tables are polished to a degree that is rarely seen at the preprint stage. It fits the scope of AMT well and should be of value for the community after some clarifications.

Major points:

1) I partially agree with reviewer #1 that the overall idea looks a lot like something that could have been achieved in a simple comparison of concentration measurements. If you assume little error in the sampling itself and that the two analysers work with practically identical samples, there would be no reason to do this in the field, to generate the increased $N_2O$ concentrations via fertilisation instead of using standards, or to even calculate the fluxes at all (which are of course identical if the concentrations are identical). For an instrument comparison these would all be unwanted potential sources of error and confounding variables in the analysis.

That said, I think I see the authors' reasoning, which is to showcase that their idea actually works well in practice and for its intended purpose (measuring fluxes). It is an unfortunate truth that just because something works well in the laboratory doesn't necessarily mean that it must work well in the field.

I think this misconception is something that can be remedied quite easily by explicitly discussing early in the manuscript how and why this is much more than just comparing two instruments' ability to measure concentrations. It left me quite puzzled throughout half of the manuscript, because it only really becomes clear after reading and thinking about it for a while.

2) I don't really get the data workflow. What does the QCL output (shouldn't it directly be ppb?), what is QCL peak area in mV supposed to be and how was it translated into concentrations? Figure S1 also doesn't make a lot of sense to me due to this. What is "$N_2O$ calculated"? If you did a calibration with standards in glass syringes and those result in higher peak areas, how can you then accurately calculate concentrations for samples that were obtained with plastic syringes which apparently result in lower peak

areas...? Obviously it did work in some way or you wouldn't get so similar results to GC, but you have absolutely lost me somewhere on the way there. Expanding section 2.3 would help a lot. I would like to see basically a recipe to get from the QCL output to whatever you did in Fig. S1 (and further)

3) This is in regards to L327-339 - I fully admit I'm not overly familiar with bioequivalence statistics, but I have a strong feeling that you are boldly overstating what it can do. From what you wrote and what I could find in your sources, it's still just frequentist inferential statistics.

Don't get me wrong, I applaud that you are willing to do solid statistics and think outside the old t-tests-and-scatterplots box. But to me this honestly just looks like another type of null-hypothesis significance testing, with even more 100 % arbitrary (but hopefully consensus-based) thresholds and ranges. I.e. there is nothing objective and certainly nothing that justifies calling something a "proof" about it.

I suggest to word pretty much everything about bioequivalence with a bit less praise. It's a good and interesting approach and it makes sense to apply it to fluxes, but that's about it.

4) In section 3.4.2 you write about "using a QCL [...] without much disruption of other measurements". I respectfully disagree with that, consdering that in section 3.3 you say that you already need an initial lag time of 10 to 30 mins, so I assume you lose at least two half-hourly EC measurements for a single sample (How much is it actually? Please state it in the manuscript!). This is something that I, as someone working at an EC station, would not want to sacrifice at least during daytime and/or after significant management events (but I wouldn't want to inject gas samples the whole night either). You mention postponing analysis later in section 3.4.2. I would like to ask you to elaborate on this idea. Would it make sense to collect multiple samples and analyse them in one batch? Is this feasible in the field? In Tab. 1 you state that you can inject 200 samples per hour. For how long can the samples be stored on site (e.g. could it wait

until the next maintenance of the EC system excl. the QCL)? I would like to see some of your ideas on this and maybe an actual example for a sampling plan that minimises EC downtime. This disruption is a core issue for anyone doing EC, so it should play a much more central role in the discussion.

Minor points: * L97: Please give a justification for the very high application rates somewhere around here. * Check typographical rules for formulae, etc. Variables should be cursive (but _descriptive_ indices upright). * L163: "Since the quadratic fit suited lower C_N2O better than a linear fit, quadratic models were preferred [...]" The fit will naturally be better (in terms of $R^2$) if you throw more parameters at your model. Am I missing something here? * L203: Power depends on (among other things) the sample size. You can't just say a 90 % CI corresponds to 80 % statistical power. * L268: I think here you can replace "might explain" with "explains". At least to my understanding it's somewhat trivial that you calculate a larger flux if you measure higher concentrations with the QCL, no? * L314-315: Have you tested injecting blanks and see what happens? * L520: Typo "TAylor" * Figure 3: Panel c and d should have equal scaling on their respective x and y axes (i.e. the 1:1 line should be the diagonal). * Table S3: Where were the soil samples taken?

---

## Author Comment (AC2) · 6 Aug 2020

*We, first of all, like to thank Referee #2. In fact, we were delighted to receive a thoroughly reviewed and thought-through feedback, which is contributing to the quality of our manuscript. Our response to the comments made by Referee #2 is indicated with an Asterix (*) at the beginning and the end of each comment:*

In their submission "A novel injection technique: using a field-based quantum cascade laser for the analysis of gas samples derived from static chambers" Wecking et al. report on their experiences with a sampling strategy that analyses both eddy covariance data and chamber measurements on-site sing the same instrument. They calculated

fluxes of N2O from concentrations measured by GC (conventional) and directly injected into a QCL (novel) in three different fertilisation treatments (+ control). Both concentrations and fluxes were analysed with a variety of statistical methods and found to be practically equivalent, with the exception of near-zero flux conditions. I have thoroughly enjoyed reviewing this manuscript. It is well-structured, well-written, and the figures and tables are polished to a degree that is rarely seen at the preprint stage. It fits the scope of AMT well and should be of value for the community after some clarifications.

Major points: 1) I partially agree with reviewer #1 that the overall idea looks a lot like something that could have been achieved in a simple comparison of concentration measurements. If you assume little error in the sampling itself and that the two analysers work with practically identical samples, there would be no reason to do this in the field, to generate the increased N2O concentrations via fertilisation instead of using standards, or to even calculate the fluxes at all (which are of course identical if the concentrations are identical). For an instrument comparison these would all be unwanted potential sources of error and confounding variables in the analysis. That said, I think I see the authors' reasoning, which is to showcase that their idea actually works well in practice and for its intended purpose (measuring fluxes). It is an unfortunate truth that just because something works well in the laboratory doesn't necessarily mean that it must work well in the field. I think this misconception is something that can be remedied quite easily by explicitly discussing early in the manuscript how and why this is much more than just comparing two instruments' ability to measure concentrations. It left me quite puzzled throughout half of the manuscript, because it only really becomes clear after reading and thinking about it for a while.

*As pointed out by Referee #2, it was our intended purpose to test whether a QCL analyser could be used for the injection of chamber derived N2O samples in the fields. We consciously decided to develop a sampling/analytic approach that would reliably work in the fields, instead, of testing a laboratory approach only. Based on own experiences, we found that an applied approach as such would be very useful for other users

who conduct measurements with static chambers and eddy covariance but are interested in shortening analysis times, receive immediate measurement results and have an interest in reducing costs by accessing the full potential of the QCL analyser. We acknowledge the Referee's comment and are going to add in a specific acknowledgement of the purpose of our work at the end of the introduction in the revised version of the manuscript.*

2) I don't really get the data workflow. What does the QCL output (shouldn't it directly be ppb?), what is QCL peak area in mV supposed to be and how was it translated into concentrations? Figure S1 also doesn't make a lot of sense to me due to this. What is "N2O calculated"? If you did a calibration with standards in glass syringes and those result in higher peak areas, how can you then accurately calculate concentrations for samples that were obtained with plastic syringes which apparently result in lower peak areas...? Obviously it did work in some way or you wouldn't get so similar results to GC, but you have absolutely lost me somewhere on the way there. Expanding section 2.3 would help a lot. I would like to see basically a recipe to get from the QCL output to whatever you did in Fig. S1 (and further)

*For clarification: The injection of 1 mL sample volumes into the QCL resulted in an output of peak area data (i.e. similar to the output as received after GC analysis). In the subsequent data analysis, we calculated the area under each peak for 1) sample injections of unknown N2O concentration, and 2) injected N2O standards of known N2O concentration. We used the results from 2) to generate quadratic models which we then fitted to the data from 1) to translate outputs into N2O concentrations (see Section 2.3 of the manuscript). We agree to the referee that explaining the procedure demands additional clarification, which we will implement to L. 159 of the manuscript. We conducted preliminary tests using glass and insulin syringes in which we applied the above translation from N2O peak areas to N2O concentrations. All data presented in the manuscript were purely based on injections made using glass syringes. This applied to both: 1) the injection of N2O samples and 2) N2O standards. To avoid confusion about the use of different syringe types, we now intend to remove Fig. S1 from the supplementary material. Instead, we will include additional explanations expanding on Section 2.3 and our work-flow procedures as suggested by the referee.*

3) This is in regards to L327-339 - I fully admit I'm not overly familiar with bioequivalence statistics, but I have a strong feeling that you are boldly overstating what it can do. From what you wrote and what I could find in your sources, it's still just frequentist inferential statistics. Don't get me wrong, I applaud that you are willing to do solid statistics and think outside the old t-tests-and-scatterplots box. But to me this honestly just looks like another type of null-hypothesis significance testing, with even more 100 % arbitrary (but hopefully consensus-based) thresholds and ranges. I.e. there is nothing objective and certainly nothing that justifies calling something a "proof" about it. I suggest to word pretty much everything about bioequivalence with a bit less praise. It's a good and interesting approach and it makes sense to apply it to fluxes, but that's about it.

*The reviewer is correct that the bioequivalence test performed in our study (as described in Section 3.4.1) is a frequentist hypothesis test. However, what is vitally important here is that the test is designed to assess (e.g. that two products are the same) . In our experiment, we were interested in whether calculated N2O concentrations/fluxes from QCL (FN2O_QCL) are effectively the same as those determined by a standard method, i.e. in our case laboratory-based GC (FN2O_GC). The bioequivalence test allowed us to assess this. The equivalence range (i.e. maximum acceptable difference) does need to be specified, and this perhaps could be seen as arbitrary. However, here it is important to be aware of the following: that 1) this is set a-priori to analysis and 2) this explicitly defines what is meant by bioequivalent (as explained in Section 2.5). We have endeavoured to make very clear in the paper what the equivalence range is (e.g. L210, L334, L338, L342, L350, Figure 6) and how this determines the definition of "bioequivalence" (e.g. L205-215). Following the reviewer's advice, we are going to rephrase all sentences regarding bioequivalence so that we don't overstate what the methodology can do. This includes changing "proof of bioequivalence" to "evidence of

bioequivalence" (lines 334, 336, 340, 345, 351).*

4) In section 3.4.2 you write about "using a QCL [...] without much disruption of other measurements". I respectfully disagree with that, considering that in section 3.3 you say that you already need an initial lag time of 10 to 30 mins, so I assume you lose at least two half-hourly EC measurements for a single sample (How much is it actually? Please state it in the manuscript!). This is something that I, as someone working at an EC station, would not want to sacrifice at least during daytime and/or after significant management events (but I wouldn't want to inject gas samples the whole night either). You mention postponing analysis later in section 3.4.2. I would like to ask you to elaborate on this idea. Would it make sense to collect multiple samples and analyse them in one batch? Is this feasible in the field? In Tab. 1 you state that you can inject 200 samples per hour. For how long can the samples be stored on site (e.g. could it wait until the next maintenance of the EC system excl. the QCL)? I would like to see some of your ideas on this and maybe an actual example for a sampling plan that minimises EC downtime. This disruption is a core issue for anyone doing EC, so it should play a much more central role in the discussion.

*We acknowledge the Referee's interest in the capability of the QCL device to provide rapid analysis of chamber derived N2O samples. As pointed out in L. 133, Section 2.2.3, the conversion of the EC QCL system to the injection mode took less than 30 min. This time comprised the establishment of an operational set-up including: to assemble and mount tubes, gas bottles etc. to the QCL, adjust the flow of N2 carrier gas and let the temperature-controlled enclosure system of the QCL housing adjust and recover from the disruptions made (e.g. lid-opening, flow-rate change). Afterwards and as identified by the referee, it is recommended to inject as many samples as available and avoid EC down-time. In L. 378, we have already indicated (see quotation of Faust and Liebig, 2018) that storing gas samples in Exetainer vials is possible and allows to minimise disruptions to EC measurements. We will add a couple of sentences to the end of Section 3.4.2 to provide further clarification about the advantages of our

injection technique – as indicated by the referee.*

Minor points: - L97: Please give a justification for the very high application rates somewhere around here. - Check typographical rules for formulae, etc. Variables should be cursive (but _descriptive_ indices upright). - L163: "Since the quadratic fit suited lower $C_{N2O}$ better than a linear fit, quadratic models were preferred [...]" The fit will naturally be better (in terms of R2) if you throw more parameters at your model. Am I missing something here? - L203: Power depends on (among other things) the sample size. You can't just say a 90 % CI corresponds to 80 % statistical power. - L268: I think here you can replace "might explain" with "explains". At least to my understanding it's somewhat trivial that you calculate a larger flux if you measure higher concentrations with the QCL, no? - L314-315: Have you tested injecting blanks and see what happens? - L520: Typo "TAylor" - Figure 3: Panel c and d should have equal scaling on their respective x and y axes (i.e. the 1:1 line should be the diagonal). - Table S3: Where were the soil samples taken?

*We are going to acknowledge these minor comments in the revised version of our manuscript. A detailed description of changes made will be provided to the associated editor with re-submission of the manuscript. This description will reiterate a detailed reasoning for changes made and indicate where (line number), how and to what degree the comments from Referee #1 and Referee #2 were implemented.*

*Kind regards in the name of all authors from Hamilton, New Zealand – 06/08/2020 Anne Wecking*

FAUST, D. R., and LIEBIG, M. A.: Effects of storage time and temperature on greenhouse gas samples in Exetainer vials with chlorobutyl septa caps, MethodsX, 5, 857-864, https://doi.org/10.1016/j.mex.2018.06.016, 2018.
* * *

---

## Author Response (AR1)

**TO THE EDITOR**

Dear Associate Editor,
Dear Dr Daniela Famulari,

We are pleased to forward a revised version of our manuscript "A novel injection technique: using a field-based quantum cascade laser for the analysis of gas samples derived from static chambers" to the ATM journal.

Attached you will find our response to the comments made by referee #1 and #2 in the public discussion forum of ATM.

Referee comments are highlighted by yellow captions. Author comments and changes made to the manuscript were written in blue font with *italics* indicating the changes.

We thoroughly enjoyed the review process and believe that the referees' comments have added to the quality of our manuscript.

Kind regards in the name of all authors

Anne Wecking

**REFEREE #1**

**Reviewer comment:**
The study "A novel injection technique: using a field-based quantum cascade laser for the analysis of gas samples derived from static chambers" by Wecking et al. describes an experiment in which samples of N2O measured using the static chamber method are then analysed on a QCL and GC instrument for comparison. The study is well written and presented, but there are some over simplifications that should be addressed, in how the complexity of the system is described and the way the data is handled in the study. I advise quite a significant re-write focussing on the actual focus of the study, which is how the instruments compare.

*Author comment:*
*We like to thank Referee #1 for her/his extensive comments.*
*In response to the above: Our interest was not only to compare two different analytic devices but to further the suitability of a field-based quantum cascade laser (QCL) and associated sampling/measurement procedures to measure chamber derived samples of nitrous oxide (N2O). Of particular interest for us was to develop a technique that is applicable and easy to conduct in field environments, and that can be used in conjunction with eddy covariance (EC). Furthermore, we compared resource efficiency of the whole operation which is critical for decisions that will be made by researchers when considering actual costs of different research approaches and analytical devices.*

**Reviewer comment:**
My first comment is that this study is essentially a comparison of concentrations measured using two instruments. This work could have been carried out with gas standards without the need for any chamber measurements. The real point of the study here is whether gas injected through a QCL following this setup is a valid way to measure gas concentrations. If so, then fluxes calculated from the sample will compare well regardless.

Author comment:
Using gas standards would have allowed for a comparison between GC and QCL. However, this would not have included the whole sampling/analysis approach. Initial lab-based comparisons of QCL to other analytic devices have already been provided in the literature and shown that the suitability of a

QCL device to quantify N2O standards is without doubt (Zellweger et al. (2019); Rosenstock et al. (2013) ). We were, therefore, highly interested in whether the lab-based accuracy and precision of a QCL could also be achieved in a field-based "mini-lab" – and under realistic sampling procedures that included N2O sampling from static chambers, sample separation to inject identical sample volumes and concentrations into either of the two analysers, sample storage, sample and data processing.

**Reviewer comment:**
The novelty of the method is that the authors get past the inability of the QCL to measure actual concentrations of a sample by using standards to integrate peak areas, similar to the way the GC reports measurements, thus reducing the sample volume required.

Author comment:
Agreed, this is one advantage of using the QCL in the field setting.
Further benefits were 1) the process of getting to the point of receiving these peak area data (e.g. developing sampling procedures and the injection set-up in a field environment); and 2) to test whether the QCL could reliably process samples of low, medium and high N2O concentration (that were derived from static chambers in a real scenario including method specific uncertainties).

**Reviewer comment:**
I'd like to see some examples of the QCL concentration output at 10Hz while measuring low, medium and high concentrations of N2O in the standards to observe the shape of the peaks that are integrated. I think the presentation of the integration of these peaks far outweighs the flux work. By the authors own numbers, I believe a flush rate of the cell is greater than 1 second, so I'd be interested to see how the laser reports concentrations while measuring at 10 Hz, and what the noise looks like.

Author comment:
L. 150: All our injections were conducted at a 10 Hz frequency, which means that all QCL concentration data reported in the manuscript were consistently determined at this particular frequency. Examples of how low to high N2O concentrations were reported by the QCL were provided in the supplementary material of the work, i.e. Figure S1 a) and b). Raw data can also be found in the data repository associated with the manuscript. However, we are in line with the Referee that including a raw 10 Hz time series of injection peaks might be a useful addition to the manuscript/the supplementary material and will be considered.

Changes made to the manuscript:
**Supplementary material** We addressed the referee's #1 interest in our data processing procedures by introducing a new figure to the supplements. The new Figure (named Figure S1) shows the raw QCL output data (peak areas) that we received after manual injections into the QCL at different low to high N2O concentrations. The figure depicts an example of a time series of injected N2O samples and N2O standards. The shape of the peaks used for later integration is illustrated. We believe, that the new Figure S1 will visualise and complement the content of Section 2.3 nicely and will help to clarify the referees' comments (see comments made by referee #2 below, major comment 2).

**Reviewer comment:**
As this is the real novelty in the manuscript, much more emphasis should be on the outputs of the instrument itself, and less on the flux measurements.

Author comment:
The focus of our manuscript is on the concentration output of the QCL and GC analysers as represented by the majority of section 2 and section 3.2, 3.3, 3.4. Calculating the resulting N2O fluxes was a necessary exercise in addition to the emphasis on N2O concentrations. We like to point the

Referee to the objective of the bioequivalence analysis applied to both N2O concentration and flux data (see Section 3.4.1). Determining the suitability of a measurement device for a particular purpose has to be evaluated. However, a simple comparison of concentration data based on linear regression does not satisfy this purpose. Consequently, we were led by the intention to not only assess the degree to which the two methods (GC and QCL) would agree (orthogonal regression, Bland Altman). But to further determine whether N2O concentration and flux data, in fact, were statistically speaking "the same" (i.e. bioequivalent). Receiving reliable N2O fluxes when using the QCL injection technique was the ultimate goal of our study. We like to acknowledge that applying statistical tests like bioequivalence describes a concept most N2O flux researchers might be unfamiliar with. However, to assess the comparability of data derived from two different methods these tests are essential as was our evaluation of N2O flux data to discuss the relevance of our results in a real world scenario.

**Reviewer comment:**

I agree with many of the points the authors make, and I feel the study has value as a reference for people who may want to use the method presented in future studies; however, I don't see it being popular. Most Eddy C sites will value the collection of uninterrupted data over the ability to run the instrument as a makeshift GC.

Author comment:

It is good that we have general agreement about the value of the work. One of the key purposes of this work was to provide information for operators to make informed decisions about how they might operate their eddy covariance sites and consider trade-offs of data loss for short periods of time. How popular this approach becomes remains to be seen but was not really a focus of the current study. Thus, even if not prominent at present, we are convinced that using a QCL in the most cost- and time-efficient way to suit multiple research purposes provides a promising future application.

Changes made to the manuscript:

Our injection method does not greatly interfere with EC measurements. If chamber samples are collectively injected into the QCL, it becomes possible to minimise EC downtime even further. To highlight this capability of our method, we changed the following parts of the manuscript:

**L. 384** Clarification of wording to point out that the QCL needs an initial lag time of 10-30 min (assembling of the set-up) before manual injections become possible. *"Depending on the EC QCL system, an initial lag time of 10 to 30 min before injections might be required in order to assemble the operational set-up (Section 2.2.3) and ensure sufficient stabilisation of pressure and temperature in the QCL sample cell.* **L. 386** *Given a flow rate of 1 L min–1, rapid injections into the QCL become possible shortly afterwards with a delay between single injections of 1 mL sample volumes of only 5 to 8 sec."*

**L. 474 following** We addressed the referee's comment to expand on the idea to minimise EC downtime by adding the following sentences to the paragraph: *"Nonetheless, we here recommend to collectively inject a great number of N2O samples in order to minimise the downtime of EC measurements and other interferences made to the QCL. For instance, we were able to inject a total of around 700, 1 mL samples (432 samples, 268 standards) within four hours into the QCL (Table 1). Prior to QCL analysis samples had been kept in septum-sealed Exetainers that can store gas samples for up to 28 days at any temperature between -10 and 25°C (Faust and Liebig, 2018)".*

**Reviewer comment:**

Choosing manual injections over the more common auto injection systems used by the GC also introduce a bit of a time cost and add room for human error.

Author comment:
This is correct and this human error is captured within the error reported in the paper. Sample loops could be included in our proposed setup to further reduce the error, if desired. The time taken for one injection was less than 10 seconds and accounted for in our time analysis (Table S2) which shows overall a large time saving of, in our case, multiple days (considering transport of samples to the lab and analysis time, see Table 1). We could show that at a given flow rate of 1 L min-1 the delay time between single injections of 1 mL sample volumes was generally short (5 to 8 sec). The return of the N2O concentration in the QCL sample cell to basically zero (= background level of the N2 carrier gas) was straight forward when using visual examination of the real-time curve in TDL-Wintel. Suitable delay times were, thus, easily adjustable if the sample concentration would have exceeded 20 ppm N2O (line 312). In our study, field samples did not exceed a N2O concentration greater 10 ppm. This supported the applicability of our approach. It is to consider that using an auto-injection system would have to account for different delay times after sample injection and, therefore, would likely include longer sample delay and overall sample analysis times.

**Reviewer comment:**
In terms of running costs, a QCL requires an air conditioned site with mains power to operate in EC mode, which is essentially a lab in itself. This system is expensive to run in terms of power and replacement of parts (pump maintenance, laser lifetime of approx. 7 years etc.). The author's point is that this system is already up and running, so the additional ability to do chamber measurements doesn't add cost.

Author comment:
We agree.

**Reviewer comment:**
This is true, and although it's also true to say you don't need to take samples off-site, you will still likely have to travel some distance as plot experiments really shouldn't be setup within the footprint range of an eddy C system (ideally more than 1 km away due to the exponential rise in fluxes observed after N fertiliser application and the potential for advection effects which nullify the assumptions made by the eddy c method). The air conditioned site thus becomes a mini-lab, in some cases closer to a field site where chamber measurements are made. Some discussion on the limitations of use is required, as the study seems to suggest N fertiliser plots could be setup next to an EC tower which would be very unwise unless the plots were to mimic the exact conditions of the field of interest to the EC measurements.

Author comment:
We positioned and aligned all static chamber measurements with overall site management conditions and associated research goals. Chambers were placed outside the immediate footprint of the eddy flux tower as determined by Wall et al. (2020). This meant that the chamber trial area was about 100 m (walking distance from the tower) to avoid the Referee´s concerns. Our previous work (Wecking et al., 2020), provided an example of how to conduct chamber measurements near to an EC tower. We demonstrated that the flux signal originating from the chamber plots was not strong enough to impact our EC measurements. To address the reviewers concern, we are going to add in a sentence recommending that care has to be taken when locating chambers near an EC tower to avoid cross-contamination.

Changes made to the manuscript:
**L. 94, 95** We made clear that fluxes generated from the chamber plots did not nullify the assumptions made by the EC method. The following sentence was added to the manuscript: *"The physical distance between chamber plots and EC tower ensured that the EC footprint did not experience cross-contamination from chamber N2O fluxes (Wall et al., 2020)."*

**Reviewer comment:**
In terms of the flux experiment, I find the application of fertilisers to be far larger than is common practice. 300 Kg N ha-1 is very large, and 600 and 900 is beyond realistic. In these cases I assume some kind of saturation of N in the soil and N2O in the chamber during a 45 minute enclosure which would also affect the magnitude of the fluxes observed. In any case, the fluxes reported are of little use other than to compare the instruments.

Author comment:
Applications rates of ammonium nitrate fertiliser were chosen intentionally to trigger different low, medium and high N2O fluxes (line 18, line 95 f.). In this study, we did not intend to mimic realistic fertiliser application scenarios. Having said this, however, the N deposited in a single urine patch of dairy cattle (i.e. a major source of N2O emissions) are comparable to N loading commonly observed at 600 kg N2O-N ha-1 (Selbie et al., 2015, p. 238 and references therein).

Changes made to the manuscript:
We addressed this comment in **L. 116-119** *"Ammonium nitrate (AN) fertiliser was used as a treatment and applied at different rates to ensure production of a wide range of low to high $C_{N2O}$ in the chamber headspace* for subsequent flux measurements. [...] *The rates of AN applied were to match nitrogen loading commonly found in cattle excreta patches, which are the main sources of $N_2O$ in grazed pastures (Selbie et al., 2015)."*

**Reviewer comment:**
In this case, there is no reason to take means from plots. Due to the log-normal nature of N2O emissions, the (arithmetically derived) mean values reported from a small n size (less than 25 chambers) is going to be fairly uncertain. Without accounting for the log-normal nature of these fluxes in both time and space, any uncertainties in cumulative flux estimates are not statistically meaningful.

Author comment:
The study´s intention was not to discuss the effect of different applications rates of ammonium nitrate fertiliser on N2O fluxes. Treatment effects were only of secondary interest. Different rates of fertiliser were only applied to result in a wide range of N2O fluxes (low to high), and thereby to allow for a methodological comparison of GC and QCL data.

**Reviewer comment:**
I return to my original point that this comparison is of gas concentrations and not of plots. The fluxes derived from both instruments are valuable on a 1:1 basis as presented in Figs 3a and 3b. That's all the paper requires and it's a great result in terms of showing the system works as well as the GC.

Author comment:
Our study is a comparison of two methods, GC and QCL. Validation of this comparison is retrieved from different statistical tests (orthogonal regression, Bland Altman and bioequivalence) that enabled us to use a QCL for the purpose of analysing static chamber derived N2O samples under real field conditions. N2O concentration and flux data were determined and used to verify this objective.

**Reviewer comment:**
In conclusion, I think the work presented is a well carried out and valid study, but it needs a bit of a re-write to focus on the actual message, and not get distracted by flux comparisons and methods of comparing significance.

Author comment:

Perhaps this is where we disagree, we are focussed on comparing the whole process required to quantify fluxes from a chamber trial using a field-based QCL rather than a laboratory based GC. Even a field based GC would give a very different outcome because of the longer time needed for separating and analysing samples. Our study will provide a broader assessment for researchers considering using QCLs in the field including quantifying precision and accuracy of the approach coupled to time and resource needs.

In addition, testing the suitability of a new method (QCL) compared with a standard method (GC) requires suitable statistical tests that do not only compare the significance but let us evaluate the agreement and bioequivalence of these methods (line 198, we also recommend reading the references quoted in Section 2.5 – Bland and Altman, 1986 and Giavarina, 2015 – that provide powerful insights into the warrant of applying these statistical tests to both N2O concentrations and fluxes in our study).

Yours sincerely,
A. Wecking in the name of all associated Co-Authors
Hamilton, New Zealand – 30-06-2020

**REFEREE #2**

Major comments

**Reviewer comment:**
1) I partially agree with reviewer #1 that the overall idea looks a lot like something that could have been achieved in a simple comparison of concentration measurements. If you assume little error in the sampling itself and that the two analysers work with practically identical samples, there would be no reason to do this in the field, to generate the increased N2O concentrations via fertilisation instead of using standards, or to even calculate the fluxes at all (which are of course identical if the concentrations are identical). For an instrument comparison these would all be unwanted potential sources of error and confounding variables in the analysis. That said, I think I see the authors' reasoning, which is to showcase that their idea actually works well in practice and for its intended purpose (measuring fluxes). It is an unfortunate truth that just because something works well in the laboratory doesn't necessarily mean that it must work well in the field. I think this misconception is something that can be remedied quite easily by explicitly discussing early in the manuscript how and why this is much more than just comparing two instruments' ability to measure concentrations. It left me quite puzzled throughout half of the manuscript, because it only really becomes clear after reading and thinking about it for a while.

Author comment:
*As pointed out by Referee #2, it was our intended purpose to test whether a QCL analyser could be used for the injection of chamber derived $N_2O$ samples in the fields. We consciously decided to develop a sampling/analytic approach that would reliably work in the fields, instead, of testing a laboratory approach only. Based on own experiences, we found that an applied approach as such would be very useful for other users who conduct measurements with static chambers and eddy covariance but are interested in shortening analysis times, receive immediate measurement results and have an interest in reducing costs by accessing the full potential of the QCL analyser. We acknowledge the Referee's comment and are going to add in a specific acknowledgement of the purpose of our work at the end of the introduction in the revised version of the manuscript.*

Changes made to the manuscript:
**L. 79 following** We clarified the purpose of our research at the end of the introduction more

precisely: *"An important component of this comparison was to demonstrate that manual injections into the QCL offer a robust method for the use in field environments. Our analysis, therefore, reached beyond the sole comparison of two analytic devices (QCL and GC) but as well discussed the method's real world application. An evidence of concept, i.e. assessing if the injection method would result in for practical purposes equivalent N2O fluxes, was provided by statistical tests: 1) orthogonal regression; 2) Bland Altman and 3) bioequivalence analyses."*

**L. 489 and L. 516** *"practical tool"*, *"rapid field analysis of N$_2$O samples"* – We adapted our language in the conclusion of the manuscript to tie our final statements back to the last paragraph of the introduction.

**Reviewer comment:**
2) I don't really get the data workflow. What does the QCL output (shouldn't it directly be ppb?), what is QCL peak area in mV supposed to be and how was it translated into concentrations? Figure S1 also doesn't make a lot of sense to me due to this. What is "N2O calculated"? If you did a calibration with standards in glass syringes and those result in higher peak areas, how can you then accurately calculate concentrations for samples that were obtained with plastic syringes which apparently result in lower peak areas...? Obviously it did work in some way or you wouldn't get so similar results to GC, but you have absolutely lost me somewhere on the way there. Expanding section 2.3 would help a lot. I would like to see basically a recipe to get from the QCL output to whatever you did in Fig. S1 (and further)

Author comment:
*For clarification: The injection of 1 mL sample volumes into the QCL resulted in an output of peak area data (i.e. similar to the output as received after GC analysis). In the subsequent data analysis, we calculated the area under each peak for 1) sample injections of unknown N$_2$O concentration, and 2) injected N$_2$O standards of known N$_2$O concentration. We used the results from 2) to generate quadratic models which we then fitted to the data from 1) to translate outputs into N$_2$O concentrations (see Section 2.3 of the manuscript). We agree to the referee that explaining the procedure demands additional clarification, which we will implement to L. 159 of the manuscript.

We conducted preliminary tests using glass and insulin syringes in which we applied the above translation from N$_2$O peak areas to N$_2$O concentrations. All data presented in the manuscript were purely based on injections made using glass syringes. This applied to both: 1) the injection of N$_2$O samples and 2) N$_2$O standards. To avoid confusion about the use of different syringe types, we now intend to remove Fig. S1 from the supplementary material. Instead, we will include additional explanations expanding on Section 2.3 and our work-flow procedures as suggested by the referee.

Changes made to the manuscript:
We expanded on the recommendation of the referee by:

**L. 138, 139** Adding a sentence to clarify that all samples were stored in septum-seal Exetainers until the date of their analysis. *"All samples remained in the septum-sealed Exetainers until the date of their analysis."*

**L. 155** Making clear that we collectively injected all QCL samples on one day, *"17 September"*, only. The sentence in this line was modified to: "The second batch of N2O samples was analysed *collectively on the day* after *the last* chamber sampling, *17 September*, by manual injection into a continuous-wave quantum cascade laser absorption spectrometer (QCL, Aerodyne Research Inc., Billerica, MA, USA)."

**L. 182** We deleted this sentence to avoid further confusion regarding syringe types used. The deleted sentence was*: "Generally, using a 1 mL glass syringe was preferred to commonly used insulin syringes because of its higher accuracy resulting in greater output peak areas (Fig. S1c)."* We also included some new information **L. 190 following**: *"Finally, it was important to keep a record of the injected sample and standard sequence to allow for re-identification in the raw output data of the QCL."*

**Supplementary material** We re-arranged former Figure S1 and deleted part (c) from Figure S1 to avoid confusion about syringe types. Note Figure S1 is now Figure S2, since we as well added an additional figure to the supplementary material (see comment below).

**L. 195 following** We added a sentence to the first paragraph of Section 2.3 in order to highlight the importance of the relationship between peak area and $N_2O$ concentration. *"Data processing, therefore, first determined the relationship between peak area and (known) N2O concentration (CN2O) of the injected standards."* Changes in wording were also made to the first and the third sentence of this paragraph as well as to referencing the figures in the supplementary material.

**Supplementary material** To address the referee's #2 interest in our data processing procedures, we introduced a new figure to the supplements. This **new Figure S1** shows the raw QCL output data (peak areas) from manual injections into the QCL. We believe, that the new Figure S1 will visualise and complement the content of Section 2.3. (Note, former Figure S1 is now labelled as Figure S2 both in the running text and in the supplementary material. The caption of **Figure S2** has changed to: "*Tests conducted prior to the main study showing the calculated normal linear relationship between output peak area and N2O concentration (CN2O) for different scenarios and for different ranges of N2O standards injected: (a) from 0.2 to 10 ppm and (b) from 0.2 to 0.5 ppm; (c) demonstrates the effect of flow rate in L min–1 on the slope of the associated regression lines, output peak area and N2O concentration in ppm.*")

**Reviewer comment:**

3) This is in regards to L327-339 - I fully admit I'm not overly familiar with bioequivalence statistics, but I have a strong feeling that you are boldly overstating what it can do. From what you wrote and what I could find in your sources, it's still just frequentist inferential statistics. Don't get me wrong, I applaud that you are willing to do solid statistics and think outside the old t-tests-and-scatterplots box. But to me this honestly just looks like another type of null-hypothesis significance testing, with even more 100 % arbitrary (but hopefully consensus-based) thresholds and ranges. I.e. there is nothing objective and certainly nothing that justifies calling something a "proof" about it. I suggest to word pretty much everything about bioequivalence with a bit less praise. It's a good and interesting approach and it makes sense to apply it to fluxes, but that's about it.

Author comment:

*The reviewer is correct that the bioequivalence test performed in our study (as described in Section 3.4.1) is a frequentist hypothesis test. However, what is vitally important here is that the test is designed to assess (e.g. that two products are the same) . In our experiment, we were interested in whether calculated $N_2O$ concentrations/fluxes from QCL ($F_{N_2O\_QCL}$) are effectively the same as those determined by a standard method, i.e. in our case laboratory-based GC ($FN_2O_{\_GC}$). The bioequivalence test allowed us to assess this. The equivalence range (i.e. maximum acceptable difference) does need to be specified, and this perhaps could be seen as arbitrary. However, here it is important to be aware of the following: that 1) this is set a-priori to analysis and 2) this explicitly defines what is meant by bioequivalent (as explained in Section 2.5). We have endeavoured to make very clear in the paper what the equivalence range is (e.g. L210, L334, L338, L342, L350, Figure 6) and how this determines the definition of "bioequivalence" (e.g. L205-215). Following the reviewer's advice, we are going to rephrase all sentences regarding bioequivalence so that we don't overstate

what the methodology can do. This includes changing "proof of bioequivalence" to "evidence of bioequivalence" (lines 334, 336, 340, 345, 351).*

Changes made to the manuscript:
**L. 26, 27, 422, 424, 428, 431, 432, 496** We changed the word *"bioequivalence" to "equivalence"* to specify our language and indicate that we precisely define what is meant when using the term *"equivalence range"*.

**L.420, 426, 429, 434, 437, 496** To satisfy the reviewer, we changed *"proof" to "evidence"* and used a more defined language around the terminology of *"bioequivalence"*.

Reviewer comment:
4) In section 3.4.2 you write about "using a QCL [...] without much disruption of other measurements". I respectfully disagree with that, considering that in section 3.3 you say that you already need an initial lag time of 10 to 30 mins, so I assume you lose at least two half-hourly EC measurements for a single sample (How much is it actually? Please state it in the manuscript!). This is something that I, as someone working at an EC station, would not want to sacrifice at least during daytime and/or after significant management events (but I wouldn't want to inject gas samples the whole night either). You mention postponing analysis later in section 3.4.2. I would like to ask you to elaborate on this idea. Would it make sense to collect multiple samples and analyse them in one batch? Is this feasible in the field? In Tab. 1 you state that you can inject 200 samples per hour. For how long can the samples be stored on site (e.g. could it wait until the next maintenance of the EC system excl. the QCL)? I would like to see some of your ideas on this and maybe an actual example for a sampling plan that minimises EC downtime. This disruption is a core issue for anyone doing EC, so it should play a much more central role in the discussion.

Author comment:
*We acknowledge the Referee's interest in the capability of the QCL device to provide rapid analysis of chamber derived $N_2O$ samples. As pointed out in L. 133, Section 2.2.3, the conversion of the EC QCL system to the injection mode took less than 30 min. This time comprised the establishment of an operational set-up including: to assemble and mount tubes, gas bottles etc. to the QCL, adjust the flow of $N_2$ carrier gas and let the temperature-controlled enclosure system of the QCL housing adjust and recover from the disruptions made (e.g. lid-opening, flow-rate change). Afterwards and as identified by the referee, it is recommended to inject as many samples as available and avoid EC down-time. In L. 378, we have already indicated (see quotation of Faust and Liebig, 2018) that storing gas samples in Exetainer vials is possible and allows to minimise disruptions to EC measurements. We will add a couple of sentences to the end of Section 3.4.2 to provide further clarification about the advantages of our injection technique – as indicated by the referee.*

Changes made to the manuscript:
We believe that this disagreement was based on misinterpretation of some of the content discussed in Section 3.3. and Section 3.4.2. To clarify that sample analysis after injection into the QCL becomes possible within seconds (not half-hours as interpreted by referee #2) we clarified the following:

**L. 138** Adding a sentence to indicate that we stored all our chamber samples in septum-sealed Exetainers*: "All samples remained in the septum-sealed Exetainers until analysis."*

**L. 155** Specification of sentence content: *The second batch of $N_2O$ samples was analysed collectively on the day after the last chamber sampling, 17 September, […]." Underlined* words were added to the sentence. **Table S1**: We also added this information to the last column of supplementary Table S1.

**Supplementary material** We introduced a new figure, Figure S1, that provides an example of the temporal frequency (x-axis) at which we injected individual N2O samples into the QCL. Figure S1 is referenced in the running text of the manuscript in **L. 195, 389**

**L. 385 following** Clarification of wording to point out that the QCL needs an initial lag time of 10-30 min (assembling of the set-up) before manual injections become possible. *"Depending on the EC QCL system, an initial lag time of 10 to 30 min before injections might be required in order to assemble the operational set-up (Section 2.2.3) and ensure sufficient stabilisation of pressure and temperature in the QCL sample cell. Given a flow rate of 1 L min−1, rapid injections into the QCL become possible shortly afterwards with a delay between single injections of 1 mL sample volumes of only 5 to 8 sec."* Underlined words were added to the sentence.

**L. 390** new sentence: *"When observing the peak progression, for instance, it became noticeable that the injection of blanks (N₂ carrier gas) did not result in any changes of baseline flow."*

**L. 475 following** We addressed the referee's comment to expand on the idea to minimise EC downtime by adding the following sentences to the paragraph: *"Nonetheless, we here recommend to collectively inject a great number of N2O samples in order to minimise the downtime of EC measurements and other interferences made to the QCL. For instance, we were able to inject a total of around 700, 1 mL samples (432 samples, 268 standards) within four hours into the QCL (Table 1). Prior to QCL analysis samples had been kept in septum-sealed Exetainers that can store gas samples for up to 28 days at any temperature between -10 and 25°C (Faust and Liebig, 2018)"*. Other minor changes were made to the paragraph in order to tidy-up language and wording.
* * *
Minor comments of referee #2

- **Reviewer comment:**
  L97: Please give a justification for the very high application rates somewhere around here.
  *Changes made to the manuscript:* *"Ammonium nitrate (AN) fertiliser was used as a treatment and applied at different rates to ensure production of a wide range of low to high $C_{N2O}$ in the chamber headspace for subsequent flux measurements. […] The rates of AN applied were to match nitrogen loading commonly found in cattle excreta patches, which are the main sources of $N_2O$ in grazed pastures (Selbie et al., 2015)."* This was added to **L. 116-119** to address the comments of referee #2

- **Reviewer comment:**
  Check typographical rules for formulae, etc. Variables should be cursive (but _descriptive_ indices upright).
  *Changes made to the manuscript:* Equation 1 and 2 were adjusted and font changed to italics, see **L. 211** and **L. 247** of the manuscript

- **Reviewer comment:**
  L163: "Since the quadratic fit suited lower C_N2O better than a linear fit, quadratic models were preferred [...]" The fit will naturally be better (in terms of R2) if you throw more parameters at your model. Am I missing something here?
  Authors' response: We used a quadratic model to build the standard curve for the injected standards of known N2O concentration.
  *Changes made to the manuscript:* We added a new sentence to **L. 195**: *"Data processing, therefore, first had to determine the relationship between peak area and (known) $N_2O$ concentration ($C_{N2O}$) of the injected standards."* **L. 196 following** we complemented a few

sentences by adding "*standard curve*" and "*$C_{N2O}$*" to **L. 198, 202** in order to clarify our way of calculation. An addition was also made **to L. 204**, *"was between 0.3-10 ppm"* to clarify the minimum and maximum range of real sample N2O concentration.

- **Reviewer comment:**
  L203: Power depends on (among other things) the sample size. You can't just say a 90 % CI corresponds to 80 % statistical power.
  *Changes made to the manuscript:* We addressed this comment by changing "*corresponding*" to: *"(at a standard power level of 80 %)"*, **L. 263.**

- **Reviewer comment:**
  L268: I think here you can replace "might explain" with "explains". At least to my understanding it's somewhat trivial that you calculate a larger flux if you measure higher concentrations with the QCL, no?
  *Changes made to the manuscript:* We agree with the referee and changed **L. 339** from "*might explain*" to "*explain*".

- **Reviewer comment:**
  L314-315: Have you tested injecting blanks and see what happens?
  Authors' response: Yes, we have. The injection of blanks did not result in any variation of the baseline flow of N2 carrier gas.
  *Changes made to the manuscript:* We added this information to **L.390**: "*When observing the peak progression, for instance, it became noticeable that the injection of blanks ($N_2$ carrier gas) did not result in any changes of baseline flow.*"

- **Reviewer comment:**
  L520: Typo "TAylor"
  *Changes made to the manuscript:* Changed to "*Taylor*" **L. 643**

- **Reviewer comment:**
  Figure 3: Panel c and d should have equal scaling on their respective x and y axes (i.e. the 1:1 line should be the diagonal).
  Authors' response: Axis scale and orientation of Figure 3c and 3d meet the prerequisites of using and depicting an orthogonal regression correctly. This means, the slope of the orthogonal estimation line (= major axis) is intermediate between the slope of the Y. X estimation line and the inverse of the slope of the X.Y estimation line.

- **Reviewer comment:**
  Table S3: Where were the soil samples taken?
  Authors' response: As mentioned in **L. 119 following**, Section 2.2.1: *"Separate areas adjacent to the twelve chamber plots were established to collect soil samples for laboratory analyses of soil moisture and soil mineral nitrogen (Nmin). Soil moisture and water-filled pore space (WFPS) were analysed and calculated using the methods described in Wecking et al. (2020a).[…]"*

==Additional minor changes:==

Dear editor, please, note that all other (very minor) changes made to the manuscript but not specifically mentioned in the above, were implied by the authors. Additional changes made were only to: 1) correct spelling mistakes; 2) support further clarification of the content of the manuscript; and 3) indicate the placement of figures and tables in the running text of the manuscript by: [FIGURE 1 ABOUT HERE]

Also, we hope that the line numbers in the revised version of the manuscript will still match to what we have explained in the above. I.e. that there will not be a miss-match in line numbers due to any difference in the version of Microsoft Word. We used the "**show revisions in balloons**" view.

[revised manuscript text omitted]

(a) AN₀, (b) AN₃₀₀, (c) AN₆₀₀, (d) AN₉₀₀, (e)

Legend:
- ■ boundary of lower and upper 90% confidence interval (CI)
- ○ 90% CI of the difference (GC-QCL)
- bioequivalence range GC
- bioequivalence range QCL

**Figure 6:** Bioequivalence analysis for $N_2O$ concentrations ($C_{N2O}$) in (a-d) and $N_2O$ fluxes ($F_{N2O}$) in (e) with GC defined as the standard method. $C_{N2O}$ and $F_{N2O}$ based on QCL analysis were considered bioequivalent when the 90% confidence interval (CI) of the difference between QCL and GC (x-axis) was completely within the predefined ± 5% bioequivalence range of the difference of the standard method. The bioequivalence analysis was distinguished for $C_{N2O}$ by sampling interval ($t_0$ ,$t_{15}$ ,$t_{30}$ , $t_{45}$) and treatment with panel (a) showing results for control sites ($AN_0$) and panels (b), (c) and (d) for $AN_{300}$, $AN_{600}$ and $AN_{900}$ treatment sites. Similarly, a bioequivalence analysis was determined for $F_{N2O}$ in panel (e), here distinguished by AN application rate on the y-axis.

**Table 1:** Comparison of the GC and QCL injection methods. Details provided in the below table specifically relate to the application of the two techniques in this study (i.e. have not been generalised). NZD = New Zealand dollars.

| | GC | QCL |
|---|---|---|
| Capital cost per device (NZD) | 40,000 | 160,000 |
| Labour effort for preparation and data processing of 100 samples (hours) | 2 to 3 | < 1 |
| Transport of samples | required | not required |
| Storage of samples | required | optional |
| Analysis location | lab-based | field-based |
| Analysis time (days) | multiple days | immediate |
| Analysis cost per sample (NZD) | 3.5 | < 0.5 |
| Possible injections (per hour) | 7.5 | ~200 |
| Lag time between injections (sec) | 480 | < 10 |
| Injection procedure | manual/automated | manual |
| Injection of $N_2O$ standards | required | required |
| Injection volume per sample (mL) | 6 | 1 |
| Carrier gas | $N_2$ | $N_2$ |
| Flow rate (L min$^{-1}$) | 0.4 | 1 |
| Output of result data | post analysis | immediate |

**Supplementary material**

[Figure]

5   **Figure S1**: Example of QCL output data depicting how a one half-hourly peak progression sequence looked like. Panel (a) shows the full sequence for injected $N_2O$ samples and standards in a given half hour from 11-11:30 AM, 17 September 2020. Panel (b) captures three individual peaks from within this time period (1) (blue rectangle). Single measurement points are depicted by blue dots with the black line showing an interpolated curvature. Orange bars underneath individual peaks in panel (a) distinguish injected $N_2O$ standards from $N_2O$ samples. (2) identifies 1 ppm and 5 ppm standards injected after every 12

10   samples, here serving as a running control; (3) shows an example of an injected standard line of known $N_2O$ concentration (range: 0.2–10 ppm); and (4) the lag time that was required to ensure sufficient flushing of the QCL sample cell after injecting a sample or standard (here 10 ppm) of higher $N_2O$ concentration.

Figure S1

50

[Figure]

**Figure S2:** Tests conducted prior to the main study showing the calculated normal linear relationship between output peak area and N₂O concentration (C_{N2O}) for different scenarios and for different ranges of N₂O standards injected: (a) from 0.2 to 10 ppm and (b) from 0.2 to 0.5 ppm; (c) demonstrates the effect of flow rate in L min⁻¹ on the slope of the associated regression lines, output peak area and N₂O concentration in ppm.

60

**Table S1:** Chronology of experimental activities.

[revised manuscript text omitted]

**Table S5:** Bland-Altman analysis for $F_{N2O\_GC}$ and $F_{N2O\_QCL}$ distinguished by treatment in units nmol m$^{-2}$ s$^{-1}$, if not specified otherwise. This table provides a summary based on mean $F_{N2O\_GC}$ and $F_{N2O\_QCL}$ across replicates of the same treatment. Fig. 4, instead, illustrates the results of individual $F_{N2O\_GC}$ and $F_{N2O\_QCL}$ (not depicted in the below table) for each replicate and each treatment as the percentage mean difference between the two methods, i.e. GC (A) and QCL (B).

| Sampling | Treatment | GC (A) | QCL (B) | Mean | Difference | Difference (%) |
|---|---|---|---|---|---|---|
| [No.] | [kg N ha$^{-1}$] | $F_{N2O\_GC}$ | $F_{N2O\_QCL}$ | (A+B)/2 | (A-B) | ((A-B)/mean)*100 |
| 1 | 0 | 0.04 | 0.00 | 0.02 | 0.04 | 182.48 |
| 1 | 300 | 3.56 | 3.65 | 3.61 | -0.09 | -2.59 |
| 1 | 600 | 1.95 | 2.17 | 2.06 | -0.23 | -11.11 |
| 1 | 900 | 2.49 | 2.74 | 2.61 | -0.24 | -9.24 |
| 2 | 0 | 0.13 | 0.21 | 0.17 | -0.08 | -44.70 |
| 2 | 300 | 9.93 | 9.40 | 9.67 | 0.53 | 5.51 |
| 2 | 600 | 9.63 | 8.88 | 9.26 | 0.75 | 8.11 |
| 2 | 900 | 14.88 | 13.57 | 14.22 | 1.31 | 9.20 |
| 3 | 0 | 0.06 | 0.14 | 0.10 | -0.08 | -78.52 |
| 3 | 300 | 8.67 | 8.19 | 8.43 | 0.48 | 5.69 |
| 3 | 600 | 8.02 | 7.94 | 7.98 | 0.08 | 0.98 |
| 3 | 900 | 15.87 | 15.17 | 15.52 | 0.70 | 4.51 |
| 4 | 0 | 0.06 | 0.06 | 0.06 | 0.00 | 1.93 |
| 4 | 300 | 8.42 | 8.02 | 8.22 | 0.39 | 4.79 |
| 4 | 600 | 8.19 | 8.04 | 8.11 | 0.15 | 1.82 |
| 4 | 900 | 14.87 | 15.46 | 15.16 | -0.59 | -3.89 |
| 5 | 0 | -0.05 | 0.09 | 0.02 | -0.14 | -595.36 |
| 5 | 300 | 6.43 | 6.25 | 6.34 | 0.18 | 2.88 |
| 5 | 600 | 11.57 | 10.91 | 11.24 | 0.66 | 5.88 |
| 5 | 900 | 15.16 | 15.09 | 15.13 | 0.07 | 0.49 |
| 6 | 0 | 0.03 | 0.03 | 0.03 | 0.00 | 4.14 |
| 6 | 300 | 7.46 | 7.30 | 7.38 | 0.16 | 2.19 |
| 6 | 600 | 10.71 | 10.66 | 10.68 | 0.05 | 0.47 |
| 6 | 900 | 16.71 | 17.22 | 16.96 | -0.51 | -3.02 |
| 7 | 0 | 0.02 | 0.17 | 0.09 | -0.15 | -157.04 |
| 7 | 300 | 5.03 | 5.30 | 5.17 | -0.27 | -5.22 |
| 7 | 600 | 10.21 | 9.46 | 9.84 | 0.75 | 7.67 |
| 7 | 900 | 14.85 | 14.81 | 14.83 | 0.03 | 0.22 |
| 8 | 0 | 0.03 | 0.18 | 0.10 | -0.15 | -149.70 |
| 8 | 300 | 6.92 | 6.95 | 6.94 | -0.02 | -0.34 |
| 8 | 600 | 9.98 | 10.27 | 10.13 | -0.29 | -2.86 |
| 8 | 900 | 13.88 | 14.36 | 14.12 | -0.48 | -3.39 |
| 9 | 0 | 0.02 | 0.06 | 0.04 | -0.04 | -105.26 |
| 9 | 300 | 3.06 | 3.28 | 3.17 | -0.22 | -6.86 |
| 9 | 600 | 6.37 | 6.63 | 6.50 | -0.26 | -4.02 |
| 9 | 900 | 10.29 | 10.97 | 10.63 | -0.68 | -6.39 |

**Table S6:** Bioequivalence analysis for N$_2$O concentrations (C$_{N2O}$) and associated fluxes (F$_{N2O}$ in bottom panel of the table). C$_{N2O\_QCL}$ and F$_{N2O\_QCL}$ were considered bioequivalent when the 90% confidence interval of the difference was completely within the predefined ± 5% bioequivalence range of difference to C$_{N2O\_GC}$ and F$_{N2O\_GC}$ (corresponding to a test with size 0.05). rep. = replicates, d.f = degrees of freedom, s.e.d = standard error of the difference, LSD = least significant difference

| Time / Treatment | Mean | | Standard error of the difference of the mean | | | LSD | 90% confidence interval | | | Bioequivalence range | | | |
|---|---|---|---|---|---|---|---|---|---|---|---|---|---|
| | C$_{N2O\_GC}$ [ppm] | C$_{N2O\_QCL}$ [ppm] | rep. | d.f | s.e.d | s.e.d | difference (GC–QCL) | lower | upper | GC lower | GC upper | QCL lower | QCL upper |
| **AN$_0$** | | | | | | | | | | | | | |
| t$_0$ | 0.333 | 0.332 | 27 | 26 | 0.0027 | 0.0046 | 0.000 | -0.004 | 0.005 | -0.017 | 0.017 | -0.017 | 0.017 |
| t$_{15}$ | 0.333 | 0.342 | 27 | 26 | 0.0028 | 0.0048 | -0.009 | -0.013 | -0.004 | -0.017 | 0.017 | -0.017 | 0.017 |
| t$_{30}$ | 0.335 | 0.352 | 27 | 26 | 0.0029 | 0.0049 | -0.016 | -0.021 | -0.012 | -0.017 | 0.017 | -0.018 | 0.018 |
| t$_{45}$ | 0.340 | 0.354 | 27 | 26 | 0.0027 | 0.0046 | -0.014 | -0.019 | -0.009 | -0.017 | 0.017 | -0.018 | 0.018 |
| **AN$_{300}$** | | | | | | | | | | | | | |
| t$_0$ | 0.333 | 0.336 | 27 | 26 | 0.0028 | 0.0048 | -0.003 | -0.007 | 0.002 | -0.017 | 0.017 | -0.017 | 0.017 |
| t$_{15}$ | 0.822 | 0.821 | 27 | 26 | 0.1090 | 0.0186 | 0.001 | -0.017 | 0.020 | -0.041 | 0.041 | -0.041 | 0.041 |
| t$_{30}$ | 1.341 | 1.327 | 27 | 26 | 0.0168 | 0.0286 | 0.014 | -0.015 | 0.042 | -0.067 | 0.067 | -0.066 | 0.066 |
| t$_{45}$ | 1.831 | 1.804 | 27 | 26 | 0.0192 | 0.0327 | 0.026 | -0.007 | 0.059 | -0.092 | 0.092 | -0.090 | 0.090 |
| **AN$_{600}$** | | | | | | | | | | | | | |
| t$_0$ | 0.336 | 0.335 | 27 | 26 | 0.0023 | 0.0042 | 0.001 | -0.003 | 0.005 | -0.017 | 0.017 | -0.017 | 0.017 |
| t$_{15}$ | 0.912 | 0.912 | 27 | 26 | 0.0160 | 0.0273 | 0.000 | -0.027 | 0.027 | -0.046 | 0.046 | -0.046 | 0.046 |
| t$_{30}$ | 1.563 | 1.550 | 27 | 26 | 0.0242 | 0.0412 | 0.013 | -0.028 | 0.054 | -0.078 | 0.078 | -0.078 | 0.078 |
| t$_{45}$ | 2.143 | 2.104 | 27 | 26 | 0.0250 | 0.0427 | 0.039 | -0.004 | 0.082 | -0.107 | 0.107 | -0.105 | 0.105 |
| **AN$_{900}$** | | | | | | | | | | | | | |
| t$_0$ | 0.338 | 0.337 | 27 | 26 | 0.0028 | 0.0319 | 0.001 | -0.004 | 0.005 | -0.017 | 0.017 | -0.017 | 0.017 |
| t$_{15}$ | 1.285 | 1.268 | 27 | 26 | 0.0136 | 0.1380 | 0.017 | -0.006 | 0.041 | -0.064 | 0.064 | -0.063 | 0.063 |
| t$_{30}$ | 2.338 | 2.294 | 27 | 26 | 0.0325 | 0.1959 | 0.044 | -0.012 | 0.100 | -0.117 | 0.117 | -0.115 | 0.115 |
| t$_{45}$ | 3.370 | 3.379 | 27 | 26 | 0.3900 | 0.2850 | -0.009 | -0.076 | 0.058 | -0.169 | 0.169 | -0.169 | 0.169 |
| **Treatment** | F$_{N2O\_GC}$ | F$_{N2O\_QCL}$ [nmol N$_2$O m$^{-2}$ s$^{-1}$] | | | | | | | | | | | |
| AN$_0$ | 0.0387 | 0.1048 | 27 | 26 | 0.0187 | 0.0319 | -0.066 | -0.098 | -0.034 | -0.002 | 0.002 | -0.005 | 0.005 |
| AN$_{300}$ | 6.610 | 6.483 | 27 | 26 | 0.0809 | 0.1380 | 0.127 | -0.011 | 0.265 | -0.331 | 0.331 | -0.324 | 0.324 |
| AN$_{600}$ | 8.514 | 8.329 | 27 | 26 | 0.1149 | 0.1959 | 0.185 | -0.011 | 0.381 | -0.426 | 0.426 | -0.416 | 0.416 |
| AN$_{900}$ | 13.222 | 13.265 | 27 | 26 | 0.1671 | 0.2850 | -0.043 | -0.328 | 0.242 | -0.661 | 0.661 | -0.663 | 0.663 |

---

## Author Response (AR2)

**TO THE EDITOR**

Dear Associate Editor,
Dear Dr Daniela Famulari,

We like to thank you for your positive feedback and timely response after receiving the revised version of our manuscript. Please, find attached further modifications made to this revised version.

As before, editor comments are highlighted in yellow, author comments and changes made to the manuscript are depicted in blue. Note, all line numbers quoted in the revisions below refer to the "marked-up" PDF version of the manuscript.

Our particular attention was focused on improving the English language of the manuscript, which we hope has now been addressed to a level of satisfaction.

Yours sincerely in the name of all authors

Anne Wecking
* * *
**REVISIONS**

**Editor comment:**
The paper presents a study on a field application of an injection technique by means of infrared absorption spectroscopy: it focusses on the comparison of such novel application with the standard GC technique, both using static enclosures to measure N2O exchanges from the soil on the field. The scientific methodology used is good, the graphics are clear and all good standard. I think the results presented are useful for the scientific community, especially relevant to monitoring networks for non-CO2 GHG, where the usage of both micromet methods and enclosure methods is required. Some critical points were highlighted by the reviewers: the authors have subsequently provided the requested clarifications, and added specific material in the supplementary section, comments and to the main manuscript that, to my knowledge, address the raised issues. I would personally like to thank the reviewers for the careful revising work, making it easy for me to proceed with the next step without their further intervention.

Author comment: We agree. Thank you to the two referees, and the editor!
* * *
**Editor comment:**
Non-public comments to the Author: Small note: I encourage the authors to revise again the English throughout the text, and add below some language corrections that got my attention.

- L81: correct the English: "the method real-world application…"
  Author comment: The suggested change was applied.
  Changes made: " […] the method real-world application."

- L81-83: rewrite the sentence as the English is not clear.
  Author comment: The content of the sentence was rewritten and clarified as follows:
  Changes made: "Evidence of concept was provided by statistical tests to assess if the injection method would result in $F_{N2O\_QCL}$ equivalent to $F_{N2O\_GC}$, these included: 1) orthogonal regression, 2) Bland Altman, and 3) bioequivalence analyses."

- **L. 119:** "… the main source", singular.
  Author comment: Verb and object were changed to singular.
  Changes made to the manuscript: " […] which is the main source of $N_2O$ […]."

- **L198-200:** remove "Whereas". Modify after "…measured by GC analysis, however we found…"
  Author comment: The suggested change was applied, and the sentence split-up into two.
  Changes made to the manuscript: "de Klein et al. (2015) recommended the use of quadratic curves models as the standard curve for $C_{N2O}$ standards measured by GC analysis. However, we found […]."
* * *
**Further changes made to the manuscript can be found in lines…**

Manuscript: **15-18, 21, 22, 28, 31, 39, 48, 49, 52-54, 66, 72, 75, 76, 77, 104, 106, 109-111, 123, 127, 147, 149, 150, 153, 156, 158, 159, 160, 164, 167, 187, 189, 192, 199, 201, 202, 213, 216, 231, 239-244, 246, 248, 252, 255, 256, 282, 283, 285, 293, 301, 304, 317, 318, 332-339, 361, 362, 364-368, 372, 374, 382, 398-402, 410-412, 416-425, 456-462, 464-473, 484, 521-524, 528, 530, 532, 533, 536, 538, 539, 541, 542, 547, 548, 571, 577, 579-582, 584-587, 597, 599, 600, 632, 634, 643, 644, 659, 831, 840, 853, 865, 869, 875, 880, 881, 899, 912, 927, 940, 943, 954, 962,** and **963**

Supplementary material: **6, 7, 18-20, 31, 32, 41, 73**

… and address the editor's comment to provide further revision of the English language throughout the text. Adjustments made to the language focused on the use of articles ('a', 'an', 'the'), hyphens ('-') and commas (,). Redundant wording (e.g. 'for the purpose of', 'prior to') was deleted, the precision of the language enhanced, and long sentences broken into shorter sequences.

Please, note that we also re-arranged our references in the running text of the manuscript. The references should now be placed in alphabetical/chronological order. Changes apply to the following lines:

Manuscript: **49, 52-61, 65-67, 70, 73-74, 100, 208, 237, 411, 478,** and **524**

[revised manuscript text omitted]

5  **Figure S1:** Example of QCL output data depicting how a one half-hourly peak progression sequence looked like. Panel (a) shows the full sequence for injected $N_2O$ samples and standards in a given half-hour from 11-11:30 AM, 17 September 2020. Panel (b) captures three individual peaks from within this period (1) (blue rectangle). Single measurement points are depicted by blue dots with the black line showing an interpolated curvature. Orange bars underneath individual peaks in panel (a) distinguish injected $N_2O$ standards from $N_2O$ samples. (2) identifies 1 ppm and 5 ppm standards injected after every 12

10  samples, here serving as a running control; (3) shows an example of an injected standard line of known $N_2O$ concentration (range: 0.2–10 ppm); and (4) the lag time that was required to ensure sufficient flushing of the QCL sample cell after injecting a sample or standard (here 10 ppm) of higher $N_2O$ concentration.

[Figure]

**Figure S2:** Tests conducted prior to the main study showing the calculated normal linear relationship between output peak area and N$_2$O concentration (C$_{N2O}$) for different scenarios and ranges of N$_2$O standards injected: (a) from 0.2 to 10 ppm, and (b) from 0.2 to 0.5 ppm, (c) demonstrates the effect of flow rate in L min$^{-1}$ on the slope of the associated regression lines, output peak area and measured N$_2$O concentration in ppm.

25    **Table S1:** Chronology of experimental activities.

| Date | Activity |
|------|----------|
| 15-Aug-19 | Trial site fenced off |
| | Preliminary injection into QCL: testing different syringe types |
| 20-Aug-19 | Installation of chamber collars |
| 30-Aug-19 | Preliminary injections into QCL: testing different flow rates |
| 10-Sep-19 | Treatment application to chamber and soil plots |
| | Gas and soil sampling – run 1 |
| 11-Sep-19 | Gas and soil sampling – run 2 |
| 12-Sep-19 | Gas and soil sampling – run 3 & 4 |
| 13-Sep-19 | Gas and soil sampling – run 5 |
| 14-Sep-19 | Gas and soil sampling – run 6 |
| 15-Sep-19 | Gas and soil sampling – run 7 & 8 |
| 16-Sep-19 | Gas and soil sampling – run 9 |
| 17-Sep-19 | Sample injection into QCL |

**Table S2:** Certified $N_2O$ standards used in this study and associated uncertainty levels. Standards printed in bolt fond were used in quadratic curve models to calculate final $N_2O$ concentration of the samples taken from static chambers.

[revised manuscript text omitted]

**Soil nitrate** [kg $NO_3^-$ $ha^{-1}$]

| | | | | | | | | |
|---|---|---|---|---|---|---|---|---|
| 10-Sep-2019 | 2.99 | 0.37 | 83.67 | 3.87 | 104.95 | 1.33 | 267.77 | 15.17 |
| 11-Sep-2019 | 2.46 | 0.18 | 69.08 | 6.54 | 149.95 | 8.62 | 248.89 | 33.69 |
| 12-Sep-2019 | 2.29 | 0.07 | 79.41 | 6.57 | 142.52 | 8.61 | 230.94 | 7.36 |
| 13-Sep-2019 | 1.64 | 0.20 | 82.21 | 7.92 | 149.85 | 6.25 | 232.40 | 13.77 |
| 14-Sep-2019 | 1.84 | 0.35 | 73.37 | 12.71 | 114.20 | 8.41 | 237.77 | 8.96 |
| 15-Sep-2019 | 2.47 | 0.31 | 78.91 | 1.51 | 162.60 | 8.72 | 231.51 | 16.94 |
| 16-Sep-2019 | 1.85 | 0.22 | 92.49 | 16.22 | 134.38 | 7.60 | 211.88 | 18.92 |

\* flux measurements conducted twice daily at 10 AM and 12 PM

SEM = standard error of the mean

50

55

60

65

70

**Table S4:** Results from the linear functional relationship analysis (orthogonal regression). Columns labelled $C_{N2O}$ show the results of the regression analysis when using standardised $N_2O$ concentrations. Columns labelled $F_{N2O}$ provide results based on standardised $N_2O$ fluxes. Part of the regression analysis was to characterise both data streams by treatment and control, i.e. first including all data ($AN_0$, $AN_{300}$, $AN_{600}$, $AN_{900}$) in the analysis and then, separately, only the control ($AN_0$).

| | $C_{N2O}$ all AN | $C_{N2O}$ $AN_0$ only | $F_{N2O}$ all AN | $F_{N2O}$ $AN_0$ only |
|---|---|---|---|---|
| Number of observations | 432 | 108 | 108 | 27 |
| Response mean | -0.003164 | 0.3272 | -0.004008 | 0.3776 |
| Explanatory mean | 0.003164 | -0.3272 | 0.004008 | -0.3776 |
| Response variance | 0.9811 | 1.238 | 0.9860 | 1.139 |
| Explanatory variance | 1.021 | 0.5551 | 1.023 | 0.6029 |
| $r^2$ value | 0.9928 | 0.1753 | 0.9922 | 0.0939 |
| r value | 0.9964 | 0.4187 | 0.9961 | 0.3064 |
| Angle between Y on X and X on Y | 0.2068 | 42.32 | 0.2229 | 54.59 |
| Major eigenvalue | 1.999 | 1.384 | 2.005 | 1.241 |
| Minor eigenvalue | 0.003606 | 0.4096 | 0.003901 | 0.5017 |
| Bootstrap resampling | 200 | 200 | 200 | 200 |
| *Ordinary least squares:* | | | | |
| Constant | -0.006253 | 0.532 | -0.007926 | 0.537 |
| Standard error | 0.003914 | 0.1038 | 0.007861 | 0.26 |
| Lower | -0.01331 | 0.3101 | -0.02204 | -0.02 |
| Upper | 0.001710 | 0.734 | 0.006998 | 1.030 |
| Slope | 0.9766 | 0.625 | 0.9778 | 0.421 |
| *Inverse least squares:* | | | | |
| Constant | -0.006276 | 1.49 | -0.007957 | 2.072 |
| Standard error | 0.003902 | 0.6585 | 0.007902 | 82.46 |
| Lower | -0.01369 | 0.9211 | -0.02246 | -44.95 |
| Upper | 0.001786 | 3.478 | 0.007118 | 18.732 |
| Slope | 0.9837 | 3.567 | 0.9854 | 4.486 |
| *Major axis:* | | | | |
| Constant | -0.006264 | 1.108 | -0.007941 | 1.326 |
| Standard error | 0.003904 | 0.44 | 0.007872 | 40.17 |
| Lower | -0.01349 | 0.7105 | -0.02217 | -19.84 |
| Upper | 0.001610 | 2.484 | 0.006920 | 9.937 |
| Slope | 0.9801 | 2.387 | 0.9815 | 2.511 |

80

**Table S5:** Bland-Altman analysis for $F_{N2O\_GC}$ and $F_{N2O\_QCL}$ distinguished by treatment in units nmol $m^{-2}$ $s^{-1}$, if not specified otherwise. This table provides a summary based on mean $F_{N2O\_GC}$ and $F_{N2O\_QCL}$ across replicates of the same treatment. Fig. 4, instead, illustrates the results of individual $F_{N2O\_GC}$ and $F_{N2O\_QCL}$ (not depicted in the below table) for each replicate and each treatment as the percentage mean difference between the two methods, i.e. GC (A) and QCL (B).

85

| Sampling | Treatment | GC (A) | QCL (B) | Mean | Difference | Difference (%) |
|---|---|---|---|---|---|---|
| [No.] | [kg N ha$^{-1}$] | $F_{N2O\_GC}$ | $F_{N2O\_QCL}$ | (A+B)/2 | (A-B) | ((A-B)/mean)*100 |
| 1 | 0 | 0.04 | 0.00 | 0.02 | 0.04 | 182.48 |
| 1 | 300 | 3.56 | 3.65 | 3.61 | -0.09 | -2.59 |
| 1 | 600 | 1.95 | 2.17 | 2.06 | -0.23 | -11.11 |
| 1 | 900 | 2.49 | 2.74 | 2.61 | -0.24 | -9.24 |
| 2 | 0 | 0.13 | 0.21 | 0.17 | -0.08 | -44.70 |
| 2 | 300 | 9.93 | 9.40 | 9.67 | 0.53 | 5.51 |
| 2 | 600 | 9.63 | 8.88 | 9.26 | 0.75 | 8.11 |
| 2 | 900 | 14.88 | 13.57 | 14.22 | 1.31 | 9.20 |
| 3 | 0 | 0.06 | 0.14 | 0.10 | -0.08 | -78.52 |
| 3 | 300 | 8.67 | 8.19 | 8.43 | 0.48 | 5.69 |
| 3 | 600 | 8.02 | 7.94 | 7.98 | 0.08 | 0.98 |
| 3 | 900 | 15.87 | 15.17 | 15.52 | 0.70 | 4.51 |
| 4 | 0 | 0.06 | 0.06 | 0.06 | 0.00 | 1.93 |
| 4 | 300 | 8.42 | 8.02 | 8.22 | 0.39 | 4.79 |
| 4 | 600 | 8.19 | 8.04 | 8.11 | 0.15 | 1.82 |
| 4 | 900 | 14.87 | 15.46 | 15.16 | -0.59 | -3.89 |
| 5 | 0 | -0.05 | 0.09 | 0.02 | -0.14 | -595.36 |
| 5 | 300 | 6.43 | 6.25 | 6.34 | 0.18 | 2.88 |
| 5 | 600 | 11.57 | 10.91 | 11.24 | 0.66 | 5.88 |
| 5 | 900 | 15.16 | 15.09 | 15.13 | 0.07 | 0.49 |
| 6 | 0 | 0.03 | 0.03 | 0.03 | 0.00 | 4.14 |
| 6 | 300 | 7.46 | 7.30 | 7.38 | 0.16 | 2.19 |
| 6 | 600 | 10.71 | 10.66 | 10.68 | 0.05 | 0.47 |
| 6 | 900 | 16.71 | 17.22 | 16.96 | -0.51 | -3.02 |
| 7 | 0 | 0.02 | 0.17 | 0.09 | -0.15 | -157.04 |
| 7 | 300 | 5.03 | 5.30 | 5.17 | -0.27 | -5.22 |
| 7 | 600 | 10.21 | 9.46 | 9.84 | 0.75 | 7.67 |
| 7 | 900 | 14.85 | 14.81 | 14.83 | 0.03 | 0.22 |
| 8 | 0 | 0.03 | 0.18 | 0.10 | -0.15 | -149.70 |
| 8 | 300 | 6.92 | 6.95 | 6.94 | -0.02 | -0.34 |
| 8 | 600 | 9.98 | 10.27 | 10.13 | -0.29 | -2.86 |
| 8 | 900 | 13.88 | 14.36 | 14.12 | -0.48 | -3.39 |
| 9 | 0 | 0.02 | 0.06 | 0.04 | -0.04 | -105.26 |
| 9 | 300 | 3.06 | 3.28 | 3.17 | -0.22 | -6.86 |
| 9 | 600 | 6.37 | 6.63 | 6.50 | -0.26 | -4.02 |
| 9 | 900 | 10.29 | 10.97 | 10.63 | -0.68 | -6.39 |

**Table S6:** Bioequivalence analysis for N$_2$O concentrations (C$_{N2O}$) and associated fluxes (F$_{N2O}$ in bottom panel of the table). C$_{N2O\_QCL}$ and F$_{N2O\_QCL}$ were considered bioequivalent when the 90% confidence interval of the difference was completely within the predefined ±5% bioequivalence range of difference to C$_{N2O\_GC}$ and F$_{N2O\_GC}$ (corresponding to a test with size 0.05). rep. = replicates, d.f = degrees of freedom, s.e.d = standard error of the difference, LSD = least significant difference

| Time/ Treatment | Mean | | Standard error of the difference of the mean | | | LSD | 90% confidence interval | | | Bioequivalence range | | | |
|---|---|---|---|---|---|---|---|---|---|---|---|---|---|
| | C$_{N2O\_GC}$ [ppm] | C$_{N2O\_QCL}$ [ppm] | rep. | d.f | s.e.d | s.e.d | difference (GC-QCL) | lower | upper | GC lower | GC upper | QCL lower | QCL upper |
| **AN$_0$** | | | | | | | | | | | | | |
| t$_0$ | 0.333 | 0.332 | 27 | 26 | 0.0027 | 0.0046 | 0.000 | -0.004 | 0.005 | -0.017 | 0.017 | -0.017 | 0.017 |
| t$_{15}$ | 0.333 | 0.342 | 27 | 26 | 0.0028 | 0.0048 | -0.009 | -0.013 | -0.004 | -0.017 | 0.017 | -0.017 | 0.017 |
| t$_{30}$ | 0.335 | 0.352 | 27 | 26 | 0.0029 | 0.0049 | -0.016 | -0.021 | -0.012 | -0.017 | 0.017 | -0.018 | 0.018 |
| t$_{45}$ | 0.340 | 0.354 | 27 | 26 | 0.0027 | 0.0046 | -0.014 | -0.019 | -0.009 | -0.017 | 0.017 | -0.018 | 0.018 |
| **AN$_{300}$** | | | | | | | | | | | | | |
| t$_0$ | 0.333 | 0.336 | 27 | 26 | 0.0028 | 0.0048 | -0.003 | -0.007 | 0.002 | -0.017 | 0.017 | -0.017 | 0.017 |
| t$_{15}$ | 0.822 | 0.821 | 27 | 26 | 0.1090 | 0.0186 | 0.001 | -0.017 | 0.020 | -0.041 | 0.041 | -0.041 | 0.041 |
| t$_{30}$ | 1.341 | 1.327 | 27 | 26 | 0.0168 | 0.0286 | 0.014 | -0.015 | 0.042 | -0.067 | 0.067 | -0.066 | 0.066 |
| t$_{45}$ | 1.831 | 1.804 | 27 | 26 | 0.0192 | 0.0327 | 0.026 | -0.007 | 0.059 | -0.092 | 0.092 | -0.090 | 0.090 |
| **AN$_{600}$** | | | | | | | | | | | | | |
| t$_0$ | 0.336 | 0.335 | 27 | 26 | 0.0023 | 0.0042 | 0.001 | -0.003 | 0.005 | -0.017 | 0.017 | -0.017 | 0.017 |
| t$_{15}$ | 0.912 | 0.912 | 27 | 26 | 0.0160 | 0.0273 | 0.000 | -0.027 | 0.027 | -0.046 | 0.046 | -0.046 | 0.046 |
| t$_{30}$ | 1.563 | 1.550 | 27 | 26 | 0.0242 | 0.0412 | 0.013 | -0.028 | 0.054 | -0.078 | 0.078 | -0.078 | 0.078 |
| t$_{45}$ | 2.143 | 2.104 | 27 | 26 | 0.0250 | 0.0427 | 0.039 | -0.004 | 0.082 | -0.107 | 0.107 | -0.105 | 0.105 |
| **AN$_{900}$** | | | | | | | | | | | | | |
| t$_0$ | 0.338 | 0.337 | 27 | 26 | 0.0028 | 0.0319 | 0.001 | -0.004 | 0.005 | -0.017 | 0.017 | -0.017 | 0.017 |
| t$_{15}$ | 1.285 | 1.268 | 27 | 26 | 0.0136 | 0.1380 | 0.017 | -0.006 | 0.041 | -0.064 | 0.064 | -0.063 | 0.063 |
| t$_{30}$ | 2.338 | 2.294 | 27 | 26 | 0.0325 | 0.1959 | 0.044 | -0.012 | 0.100 | -0.117 | 0.117 | -0.115 | 0.115 |
| t$_{45}$ | 3.370 | 3.379 | 27 | 26 | 0.3900 | 0.2850 | -0.009 | -0.076 | 0.058 | -0.169 | 0.169 | -0.169 | 0.169 |
| **Treatment** | F$_{N2O\_GC}$ [nmol N$_2$O m$^{-2}$ s$^{-1}$] | F$_{N2O\_QCL}$ | | | | | | | | | | | |
| AN$_0$ | 0.0387 | 0.1048 | 27 | 26 | 0.0187 | 0.0319 | -0.066 | -0.098 | -0.034 | -0.002 | 0.002 | -0.005 | 0.005 |
| AN$_{300}$ | 6.610 | 6.483 | 27 | 26 | 0.0809 | 0.1380 | 0.127 | -0.011 | 0.265 | -0.331 | 0.331 | -0.324 | 0.324 |
| AN$_{600}$ | 8.514 | 8.329 | 27 | 26 | 0.1149 | 0.1959 | 0.185 | -0.011 | 0.381 | -0.426 | 0.426 | -0.416 | 0.416 |
| AN$_{900}$ | 13.222 | 13.265 | 27 | 26 | 0.1671 | 0.2850 | -0.043 | -0.328 | 0.242 | -0.661 | 0.661 | -0.663 | 0.663 |